# CoDy: Counterfactual Explainers for Dynamic Graphs

Zhan Qu [* 1 2 3]  Daniel Gomm [* 4 5]  Michael Färber [1 2]

## Abstract

Temporal Graph Neural Networks (TGNNs) are widely used to model dynamic systems where relationships and features evolve over time. Although TGNNs demonstrate strong predictive capabilities in these domains, their complex architectures pose significant challenges for explainability. Counterfactual explanation methods provide a promising solution by illustrating how modifications to input graphs can influence model predictions. To address this challenge, we present **CoDy**—**Co**unterfactual Explainer for **Dy**namic Graphs—a model-agnostic, instance-level explanation approach that identifies counterfactual subgraphs to interpret TGNN predictions. **CoDy** employs a search algorithm that combines Monte Carlo Tree Search with heuristic selection policies, efficiently exploring a vast search space of potential explanatory subgraphs by leveraging spatial, temporal, and local event impact information. Extensive experiments against state-of-the-art factual and counterfactual baselines demonstrate **CoDy**'s effectiveness, with improvements of 16% in AUFSC$_+$ over the strongest baseline. Our code is available at: https://github.com/daniel-gomm/CoDy

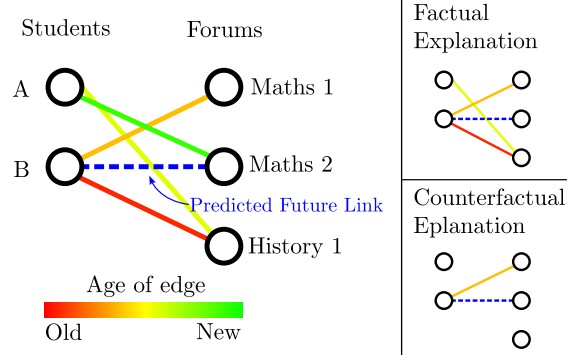

Figure 1. Bipartite dynamic graph of student posts on a university social network. Edge colors represent their ages.

## 1. Introduction

Dynamic graphs are commonly used to model applications with features that change over time, such as social networks, e-commerce platforms, and collaborative online encyclopedias like Wikipedia (Huang et al., 2024; Longa et al., 2023; Yu et al., 2023). In addition to the static topological graph structure, dynamic graphs also capture the continuously evolving temporal dependencies. Consequently, Temporal Graph Neural Networks (TGNNs) have been specifically developed to leverage the rich spatial and temporal information inherent in dynamic graphs (Zhao et al., 2019; Rossi et al., 2020; Xu et al., 2020). However, as with many deep learning models, TGNNs often function as "black boxes", offering limited insight into their decision-making processes (Xia et al., 2023; Chen & Ying, 2023).

Explainability methods for Graph Neural Networks (GNNs) aim to identify a small subset of nodes and edges that most strongly influence a model's prediction. However, most existing methods are designed for static graphs and do not generalize well to dynamic graphs, which involve complex temporal interactions (He et al., 2022; Xia et al., 2023; Chen & Ying, 2023). In temporal graphs, multiple events may occur at the same timestamp and the same position, adding complexity to the dependencies between interactions (Han et al., 2020; He et al., 2022). The explanations for TGNNs should be temporally approximate and spatially adjacent to the target (Kovanen et al., 2011; Chen & Ying, 2023).

The few methods targeting TGNN models focus primarily on factual explanations, which identify specific nodes and edges contributing to a prediction (Xia et al., 2023; Chen & Ying, 2023). They fall short of exploring how changes in the input graph could lead to different outcomes. Counterfactual explanations, which illustrate how altering the graph changes the prediction, are gaining popularity for their ability to establish causal relationships and highlight decision

---
[*]Equal contribution  [1]TU Dresden, Dresden, Germany [2]ScaDS.AI, Dresden, Germany [3]Karlsruhe Institute of Technology, Karlsruhe, Germany [4]University of Amsterdam, Amsterdam, Netherlands [5]Centrum Wiskunde en Informatica, Amsterdam, Netherlands. Correspondence to: Daniel Gomm <daniel.gomm@cwi.nl>, Zhan Qu <zhan.qu@tu-dresden.de>.

*Proceedings of the 42nd International Conference on Machine Learning*, Vancouver, Canada. PMLR 267, 2025. Copyright 2025 by the author(s).

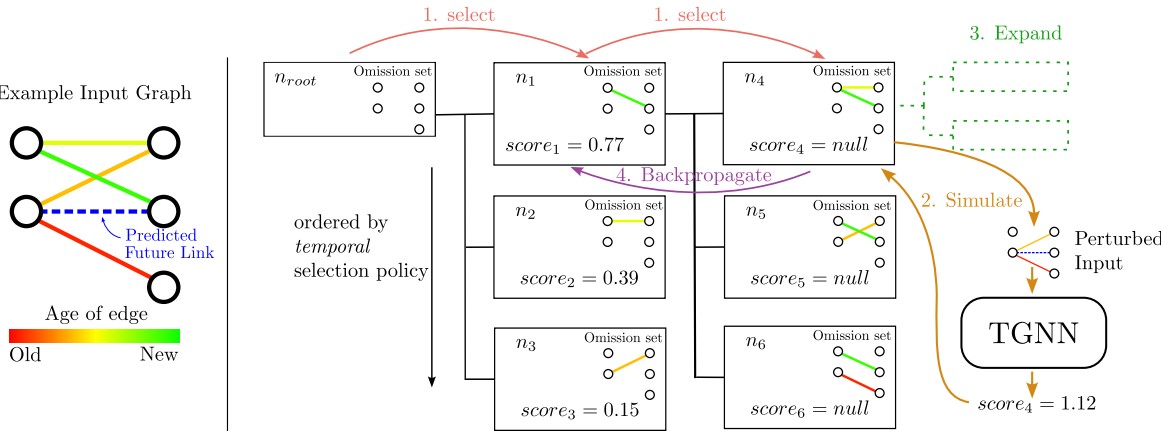

*Figure 2.* CoDy search framework. The left side shows an example of a temporal input graph, where the explained future link is highlighted in blue, and the other links are color-coded based on their ages. On the right, one iteration of the CoDy algorithm is illustrated. Each rectangle represents a node in the search tree, corresponding to a specific perturbation of the input graph.

boundaries (Byrne, 2019; Prado-Romero et al., 2024). They provide actionable insights, help identify biases, and can uncover adversarial examples (Lucic et al., 2022). Despite temporal relationships in TGNNs often naturally suggesting causal links between nodes, no existing methods effectively leverage counterfactual explanations. Fig. 1 shows an example comparing factual and counterfactual explanations for a TGNN model's prediction that Student B will post on the Math 2 forum in the future. The factual explanation traces this prediction through indirect connections: Student B previously posted on Math 1, Student A posted on both Math 1 and Math 2, and both A and B posted on History 1. This can be complex and cognitively demanding to interpret. In contrast, the counterfactual explanation simplifies the reasoning by showing that, if Student B had not posted on Math 1, the post on Math 2 would not have occurred—highlighting the minimal set of actions necessary for the prediction (Lucic et al., 2022; Prado-Romero et al., 2024). In summary, factual explanations aim to identify a subgraph with *sufficient* information to reproduce the same prediction, while counterfactual methods seek the subset of information *necessary* to alter the outcome (Tan et al., 2022).

To address the need for concise and actionable explanations, we introduce **CoDy** (**Co**unterfactual Explainer for **Dy**namic Graphs), a model-agnostic instance-level explainer for TGNNs. CoDy adopts principles from Monte Carlo Tree Search to efficiently identify counterfactual examples by modifying a subset of past events to alter the model's prediction. Further, we develop policies that leverage spatio-temporal structure and local event impacts to enhance search efficiency and effectiveness. Since CoDy is the first counterfactual explanation method for TGNNs, we develop **GreeDy** (**Gree**dy Explainer for **Dy**namic Graphs), a strong counterfactual baseline that employs greedy search.

We propose a comprehensive evaluation framework that jointly assesses factual and counterfactual explanations on dynamic graphs. Our evaluation shows that CoDy excels in generating counterfactual explanations for TGNN models such as TGN (Rossi et al., 2020) and TGAT (Xu et al., 2020), outperforming GreeDy and factual methods, including PG-Explainer (Luo et al., 2020) and T-GNNExplainer (Xia et al., 2023) across multiple datasets. Additionally, we demonstrate that incorporating spatio-temporal context and local event impact significantly improves the relevance and fidelity of counterfactual explanations. This insight opens new directions for advancing TGNN explainability by leveraging richer temporal and structural cues.

In summary, our main contributions are as follows:

1. We introduce CoDy, the first method for generating counterfactual explanations for TGNNs.

2. We develop GreeDy, a baseline approach for counterfactual explanations in dynamic graphs.

3. We create an evaluation framework tailored to counterfactual explanations on dynamic graphs.

4. We conduct extensive benchmarks of CoDy, showing its superior performance compared to counterfactual and factual baselines.

## 2. Related Work

Numerous explainability methods have been proposed for GNNs on static graphs, leveraging techniques such as gradients (Baldassarre & Azizpour, 2019; Pope et al., 2019), perturbations (Ying et al., 2019; Luo et al., 2020), decomposition rules (Schnake et al., 2021), surrogate models (Huang et al., 2022; Vu & Thai, 2020), Monte Carlo Tree Search

(MCTS) (Yuan et al., 2021), and generative models (Yuan et al., 2020; Shan et al., 2021; Miao et al., 2022; Li et al., 2023). Still, explainability for Temporal Graph Neural Networks (TGNNs) remains underexplored and challenging.

Dynamic graphs are commonly represented using Discrete-Time Dynamic Graphs (DTDGs) and Continuous-Time Dynamic Graphs (CTDGs) (Kazemi et al., 2020). DTDGs represent a system through a sequence of static snapshots, each corresponding to a specific time interval. Most existing explainability methods focus on DTDGs, relying on techniques such as surrogate models (He et al., 2022), temporal decomposition (Liu et al., 2023), and model-specific feature analysis (Fan et al., 2021). These approaches typically aggregate feature importances across snapshots to provide explanations. Recently, generative approaches such as GRACIE (Prenkaj et al., 2024) have been proposed for counterfactual explanation in DTDGs. GRACIE leverages class-specific variational autoencoders to account for distributional shifts across discrete time steps and performs generative classification by modeling class-conditional graph distributions. Predictions are inferred via latent reconstruction loss, enabling principled counterfactual generation without relying on fixed decision boundaries. However, GRACIE is tailored to discrete-time snapshots and cannot be readily applied to CTDGs without substantial adaptation to account for continuous, event-based graph dynamics.

In contrast, CTDGs capture continuously evolving interactions through timestamped events, making them more representative of real-world applications such as social networks. Despite their significance, explainability for CTDGs remains relatively unexplored, with only two methods proposed: T-GNNExplainer (Xia et al., 2023) and TempME (Chen & Ying, 2023). T-GNNExplainer employs a search-based perturbation approach, utilizing MCTS and a Multi-Layer Perceptron-based navigator to predict event importances and guide search space exploration. TempME, on the other hand, identifies key temporal motifs—recurring patterns in dynamic graphs—to explain predictions.

Existing methods, however, are limited to factual explanations that identify influential factors but do not consider alternative scenarios. Counterfactual explanations, which have been successfully applied to static GNNs (Tan et al., 2022; Lucic et al., 2022), provide a more intuitive way to explore "what if" scenarios by identifying minimal changes needed to alter model predictions. However, adapting counterfactual explanations to temporal graphs is challenging due to their evolving nature and complex dependencies.

To address this, we propose a novel counterfactual explanation approach for CTDG-based TGNNs that captures both temporal dependencies and spatial structures. Our method enhances interpretability by providing explanations that are not only intuitive but also actionable.

## 3. Preliminaries and Problem Formulation

**Continuous Time Dynamic Graphs (CTDG)**   We denote a CTDG as a sequence of timestamped events $\mathcal{G} = \{\varepsilon_1, \varepsilon_2, ...\}$. Each event $\varepsilon_i$ is associated with a timestamp $t_i$ so that $t_{i-1} < t_i < t_{i+1}$. Events represent the addition, removal, or attribute change of nodes or edges. Within a CTDG, we distinguish **temporal distance**, which is the difference in timestamps between two events, and **spatial distance**, defined as the shortest path distance within the graph structure between the nodes or edges involved in the events at their respective times. The $k$-**hop neighborhood** of an event $\varepsilon_i$ covers the set of events that occurred within $k$-edges of any nodes involved in event $\varepsilon_i$ until time $t_i$.

**Future Link Prediction**   A future link prediction model $f : \{\mathcal{G}(t_i), \varepsilon_i\} \to \mathbb{R}$ estimates the likelihood that a future event $\varepsilon_i$ will occur in the graph. Without loss of generality, we assume that the model $f$ outputs logit values as an approximation of these odds. The prediction is based on the history of events $\mathcal{G}(t_i) = \{\varepsilon_l | \varepsilon_l \in \mathcal{G}, t_l \leq t_i\}$ in the temporal graph $\mathcal{G}$ up to time $t_i$. The model $f$ can be any suitable function, such as a TGNN trained for this task. A binary classification function $p : \mathbb{R} \to \{0, 1\}$ transforms the output of $f$ into a definitive prediction, where $1$ indicates that the future link is predicted to occur.

**Counterfactual Examples in Future Link Prediction**
Counterfactual explanations provide actionable insights into specific predictions, making them particularly valuable for interpreting the outputs of complex deep graph models like TGNNs. A counterfactual example $\mathcal{X}_{\varepsilon_i}$, explaining a prediction of a future link $\varepsilon_i$, consists of a critical subset of past events $\mathcal{X}_{\varepsilon_i} \subseteq \mathcal{G}(t_i)$ necessary for the original prediction. This necessity is defined by the condition:

$$p(f(\mathcal{G}(t_i), \varepsilon_i)) \neq p(f((\mathcal{G}(t_i) \setminus \mathcal{X}_{\varepsilon_i}), \varepsilon_i)) \qquad (1)$$

For any given future link $\varepsilon_i$, there may be multiple or no counterfactual examples. A counterfactual explainer, denoted as $ex(\cdot)$, provides a rationale for the model's prediction. For future link predictions, $ex(\cdot)$ is a function that takes as input the TGNN model $f$, the temporal graph $\mathcal{G}(t_i)$, and the future link $\varepsilon_i$, and outputs a subset of the original graph's events as counterfactual explanation, i.e. any combination of past events:

$$ex : \{f, \mathcal{G}(t_i), \varepsilon_i\} \to \bigcup_{k=0}^{|(\mathcal{G}(t_i) \setminus \varepsilon_i)|} \binom{\mathcal{G}(t_i) \setminus \varepsilon_i}{k} \qquad (2)$$

The output of $ex$ can be any combination of past events. Formula 2 adheres to the notation for permutations described by Stanley (1986).

**Objectives for Counterfactual Explanation** We identify two primary objectives: maximizing the discovery of counterfactual examples and minimizing the complexity of these explanations. The goal of maximizing discoveries is to provide counterfactual explanations for as many instances as possible, while minimizing complexity aligns with the principle of Occam's razor, suggesting that explanations should be concise and contain only relevant details (Yuan et al., 2023; Tan et al., 2022). Thus, the explanation should contain as few events as possible.

## 4. Methodology

This section presents the **Co**unterfactual Explainer for Models on **Dy**namic Graphs (CoDy) alongside the baseline method, **Gree**dy Explainer for Models on **Dy**namic Graphs (GreeDy). Both approaches elucidate predictions made by TGNNs on CTDGs, focusing on efficiently navigating a defined search space to identify counterfactual examples.

**Search Space** In a dynamic graph, each subset of past events related to a future link may constitute a counterfactual example. Therefore, the complete search space $S_{\varepsilon_i}$ includes all combinations of past events of the predicted future link $\varepsilon_i$. Given that this search space grows exponentially with the number of past events, we streamline the search process by imposing spatial and temporal constraints, narrowing the focus to a subset of past events located within the spatio-temporal vicinity of the target future link event. To limit the search space spatially, we only consider events within a $k$-hop-neighborhood of the target future link. The value of $k$ is selected based on the specific dynamic graph model; for explaining TGNNs, we set $k$ equal to the number of layers in the TGNN, as contemporary TGNNs primarily aggregate information from events within this neighborhood (Yuan et al., 2021). Temporally, we retain only the $m_{max}$ most recent events that satisfy the spatial constraint. We denote this constrained subset of past events as $C(\mathcal{G}, \varepsilon_i, k, m_{max})$, yielding the constrained search space $\hat{S}\varepsilon_i$:

$$\hat{S}_{\varepsilon_i} = \bigcup_{l=0}^{|C(\mathcal{G}, \varepsilon_i, k, m_{max})|} \binom{C(\mathcal{G}, \varepsilon_i, k, m_{max})}{l} \quad (3)$$

**Search Tree Structure** The search is guided by a partial search tree $P$. Each node represents removing a unique subset of past events from the original input graph. The root node $n_{root}$ corresponds to the original input graph without removing events, while other nodes represent the result of omitting past events from the graph. The depth of a node reflects the number of omitted events, with deeper nodes corresponding to more extensive modifications.

In contrast to search trees utilized for factual explanations (Xia et al., 2023; Yuan et al., 2021), the search tree

in CoDy is specifically designed for counterfactual exploration. The goal is to identify the minimal set of events whose removal alters the model's prediction. Any child node in the tree extends the set of omitted events associated with its parent node by an additional event. This search tree is dynamically constructed as the search progresses, facilitating efficient traversal through the space of potential counterfactual explanations while minimizing unnecessary expansions. This structure promotes the identification of concise and relevant counterfactual examples, aligning with the dual objectives of maximizing discoveries and minimizing complexity.

### 4.1. Selection Policies

Selection policies guide the traversal of the partial search tree by prioritizing which nodes to explore. Each node in the search tree represents a unique combination of past events, with child nodes differing by the omission of a single event from their parent node. The selection policies rank these potential event omissions, directing the search toward efficiently discovering counterfactual examples. We propose four distinct policies:

- **Random** A baseline policy that randomly ranks events for omission from the input graph, serving as a comparison against more structured strategies.

- **Temporal** This policy ranks events based on their temporal distance from the explained event $\varepsilon_i$, assuming that more recent events have a greater influence on the model's prediction. The ranking function is defined as:

$$r_{temp}(\varepsilon_j) = |t_i - t_j| \quad (4)$$

where $t_i$ is the time of the future link event and $t_j$ is the timestamp of the past event $\varepsilon_j$. Events with smaller $r_{temp}$ (i.e., shorter temporal distance) are prioritized.

- **Spatio-Temporal** This policy integrates both spatial proximity and temporal recency. Events are initially ranked by their spatial distance to the nodes involved in the explained event $\varepsilon_i$. Among events with the same spatial proximity, they are further ranked by temporal distance. The ranking function is:

$$r_{sp-temp}(\varepsilon_j) = (d_{spatial}(\varepsilon_j, \varepsilon_i), |t_i - t_j|) \quad (5)$$

where $d_{spatial}(\varepsilon_j, \varepsilon_i)$ is the shortest path distance between the nodes of $\varepsilon_j$ and $\varepsilon_i$ at time $t_i$.

- **Event-Impact** This policy evaluates the impact of omitting each event from the full input graph on the model prediction by measuring the change in prediction logits. The event impact for an event $\varepsilon_j$ is defined as the difference between the original prediction $p_{orig}$

and the prediction $p_j$ made when $\varepsilon_j$ is removed from the input graph. The ranking is determined by:

$$\Delta(p_{orig}, p_j) = \begin{cases} p_{orig} - p_j, & \text{if } p_{orig} \geq 0 \\ p_j - p_{orig}, & \text{else} \end{cases} \quad (6)$$

The search tree is first fully expanded at depth 1, exploring the effects of separately removing each event in the search space $C(\mathcal{G}, \varepsilon_i, k, m_{max})$, calculating $\Delta(p_{orig}, p_j)$ for each event, and ranking events based on the magnitude of this change. Events with larger event impact on the prediction are prioritized.

Each selection policy offers a unique heuristic to guide the search. The Temporal and Spatio-Temporal policies leverage the structural properties of the graph by exploiting event timing and spatial relationships. In contrast, the Event-Impact policy directly measures the predictive influence of individual events, allowing for a more informed search based on the model's internal decision-making process. These policies provide a diverse set of strategies to navigate the search space efficiently and effectively.

## 4.2. GreeDy: Greedy Explainer on Dynamic Graphs

GreeDy is a search algorithm designed to find impactful counterfactual explanations by greedily exploring the search space. This exploration iteratively advances through the partial search tree $P$ by selecting paths that result in the greatest immediate change in the TGNN's prediction. Appendix A details the full algorithm.

At each iteration, GreeDy evaluates a subset of $l$ child nodes, sampled based on a selection policy. The sampled child nodes are ranked based on their potential to cause a shift in the TGNN model's prediction, with priority given to nodes that yield the most significant change. The search continues until a counterfactual example is found or further exploration no longer produces meaningful shifts.

The algorithm's output is the most impactful explanation discovered during the search, represented by the set of events whose omission causes the largest change in the model's prediction. This result can either be a valid counterfactual example or simply the set of events with the greatest non-counterfactual impact on the model's prediction.

## 4.3. CoDy: Counterfactual Explainer on Dynamic Graphs

CoDy draws on Monte Carlo Tree Search (Kocsis & Szepesvári, 2006), adapting its four key steps—Selection, Simulation, Expansion, and Backpropagation—to suit the search for counterfactual explanations in dynamic graphs. Algorithm 1 provides a high-level overview of CoDy.

Figure 2 illustrates an example iteration within the CoDy

---

**Algorithm 1** Search algorithm of CoDy.

**Input:** TGNN model $f$, input graph $\mathcal{G}$, explained event $\varepsilon_i$, selection policy $\delta$, max iterations $it_{max}$
**Output:** best explanation found
$p_{orig} \leftarrow f(\mathcal{G}(t_i), \varepsilon_i)$
$n_{root} \leftarrow (\varnothing, null, null, \varnothing, 0, null, 1)$
$it \leftarrow 0$
**while** $it < it_{max}$ and $n_{root}$ is selectable **do**
    $n_{selected} \leftarrow$ **select**$(n_{root}, \delta)$
    **simulate**$(n_{selected}, f, \mathcal{G}, \varepsilon_i)$
    **expand**$(n_{selected}, p_{orig})$
    **backpropagate**$(parent_{selected})$
    $it \leftarrow it + 1$
**end**
$n_{best} \leftarrow$ **select_best**$(n_{root})$
**return** $s_{best}$

---

algorithm. The process begins with the recursive selection of a node in the search tree. In the depicted case, node $n_1$ is initially chosen due to its high $score$. Node $n_1$ has three unexpanded child nodes, each without a score. Following the temporal selection policy, node $n_4$ is selected next. Since $n_4$ has no child nodes, the recursion halts at this point. The simulation step then follows, where the model's output is inferred by perturbing the original input graph, omitting the events corresponding to $n_4$. After this, node $n_4$ is assigned a $score$ based on the TGNN model's output. The node is then expanded to include its child nodes. Finally, the new $score$ is backpropagated through the tree, updating the scores of all parent nodes until the root node $n_{root}$. In the following subsections, we provide an overview of these steps, with full details available in Appendix B.

**Selection** Each search iteration begins by selecting a node for expansion, starting from the root and recursively traversing the tree. During this traversal, the algorithm evaluates and selects child nodes based on the selection score $sel\_score(n_k)$, which balances the need to explore new nodes and exploit known high-scoring ones:

$$sel\_score(n_k) = \alpha \cdot score_k + (1 - \alpha) \cdot score_{explore}(n_k) \quad (7)$$

The exploration score $score_{explore}(n_k)$ is derived from the 'Upper Confidence Bound 1' (Auer et al., 2002), encouraging exploration, while $score_k$ denotes the exploitation score, reflecting the node's known potential to lead to a counterfactual example based on the outcomes of previous simulations. The parameter $\alpha \in [0, 1]$ controls the trade-off between exploration and exploitation.

**Simulation** After selecting a node, the simulation step consists of inferring the prediction of the target TGNN

model using a perturbed input graph. Specifically, events associated with the selected node $n_j$ are omitted from the original input graph, resulting in a new prediction $p_j$. This prediction is then used to compute the $score_j$:

$$score_j \leftarrow \max\left(0, \frac{\Delta(p_{orig}, p_j)}{|p_{orig}|}\right) \quad (8)$$

A $score_j > 1$ signifies that the perturbation set associated with $n_j$ forms a counterfactual example. In this case, this set of nodes is saved as a counterfactual example.

**Expansion** The expansion adds child nodes to the selected node. Following the definition of the search tree, each new child node is initialized with a set of events to omit from the input graph. The attribute $score_k$ of each added child node $n_k$ is set to $null$, indicating that these nodes are unexplored.

**Backpropagation** The backpropagation recursively updates the search tree, starting from the parent node of the expanded node and proceeding until the root node. During this process, the exploitation and exploration scores are recalculated. The exploration score $score_k$ of $n_k$ is updated as:

$$score_j \leftarrow \frac{\max\left(0, \frac{\Delta(p_{orig}, p_j)}{|p_{orig}|}\right) + \sum_{n_k \in C_j}(score_k * sel_k)}{sel_j} \quad (9)$$

Here, $C_j$ refers to the child nodes of $n_j$, and $sel_j$ represents the total number of selections of node $n_j$.

**Explanation Selection** Once the search tree is fully expanded or the maximum number of iterations is reached, CoDy selects an explanation. To minimize complexity, CoDy prioritizes counterfactual examples with the minimum number of events. If multiple candidates share this minimal size, the one inducing the largest change in the TGNN's prediction (as defined in Eq. 6) is chosen, reflecting the most decisive minimal explanation. If no valid counterfactuals are found, a fallback-strategy is applied, which selects the most informative perturbation set encountered during the search, i.e., the one producing the greatest prediction shift according to Eq.6.

To enhance search efficiency, CoDy incorporates several optimization techniques: constraints are applied to minimize explanation complexity, caching mechanisms are utilized to reduce redundant calculations, and a two-stage approximation-confirmation approach is implemented to expedite inference during search.

## 5. Experiments

### 5.1. Experimental Setup

**Datasets** We evaluate on three datasets: Wikipedia (Kumar et al., 2019), UCI-Messages (Kunegis, 2013), and UCI-Forums (Kunegis, 2013). These datasets have diverse graph structures and event dynamics, allowing us to assess the generalizability of CoDy across different types of networks. The UCI datasets come from online social networks without node or edge features. Specifically, UCI-Messages is a unipartite graph of messages sent between students, while UCI-Forums is a bipartite graph of interactions between students and forums. The Wikipedia dataset consists of events representing edits made to Wikipedia articles. It is bipartite, with edges associated with attributes that detail edits to Wikipedia articles. Appendix C supplements detailed statistics and discusses the diversity of these datasets.

**Target Models** We evaluate two dynamic graph models: TGN (Rossi et al., 2020) and TGAT (Xu et al., 2020). Both are widely used in dynamic graph research (Souza et al., 2022; Chen & Ying, 2023; Xia et al., 2023) and excel at capturing temporal patterns within dynamic graphs. They are considered state-of-the-art for various dynamic graph tasks (Rossi et al., 2020; Souza et al., 2022). Despite their predictive success, TGN and TGAT operate as black-box models, providing limited insight into how input features influence their predictions. Due to page limitations, the results for TGAT are presented only in Appendix E, as they closely resemble those for TGN.

**Configurations** We train the TGN model using the "TGN-attn" configuration from the original paper (Rossi et al., 2020). The trained TGN model achieves high average precision scores (Transductive/Inductive): UCI-Messages 85.86%/83.26%; UCI-Forums 92.97%/89.39%; Wikipedia 97.80%/97.39%. The TGAT model achieves comparable performance: UCI-Messages 85.14%/82.00%; UCI-Forums 91.31%/83.73%; Wikipedia 97.47%/96.74%.

**Factual Baselines** We adapt two factual explainers as baselines: PGExplainer (Luo et al., 2020) and T-GNNExplainer (Xia et al., 2023). Though PGExplainer was originally developed for static graphs, it provides a valuable comparative perspective when adapted to dynamic contexts. We apply it with a fixed explanation sparsity of 0.2. T-GNNExplainer is designed for Temporal Graph Neural Networks and operates at the event level, making it an ideal baseline for our methodology. We follow its original specifications, using 500 iterations and 30 candidate events. While TempME (Chen & Ying, 2023) also explains TGNNs, it has a fundamentally different motif-based approach and limited comparability with our event-level counterfactual explanation framework. The authors primarily report neg-

*Table 1.* Results for the $AUFSC_+$, $AUFSC_-$, and *char* scores of different explanation methods applied to the TGN model. Results are reported for three datasets: UCI-Messages (msg.), UCI-Forums (for.), and Wikipedia (wiki.). The best result for each experimental setting is shown in **bold**, and the second best is underlined.

| | $AUFSC_+$ | | | | | | $AUFSC_-$ | | | | | | *char* | | | | | |
| | Correct | | | Incorrect | | | Correct | | | Incorrect | | | Correct | | | Incorrect | | |
| Dataset | msg. | for. | wiki. | msg. | for. | wiki. | msg. | for. | wiki. | msg. | for. | wiki. | msg. | for. | wiki. | msg. | for. | wiki. |
|---|---|---|---|---|---|---|---|---|---|---|---|---|---|---|---|---|---|---|
| PGExplainer | 0.02 | 0.03 | 0.03 | 0.08 | 0.04 | 0.11 | 0.39 | 0.35 | 0.67 | 0.61 | 0.67 | 0.54 | 0.05 | 0.07 | 0.09 | 0.17 | 0.08 | 0.22 |
| T-GNNExplainer | 0.05 | 0.03 | 0.01 | 0.14 | 0.19 | 0.10 | 0.45 | 0.36 | 0.61 | 0.53 | 0.49 | 0.43 | 0.17 | 0.08 | 0.09 | 0.39 | 0.40 | 0.34 |
| GreeDy-*rand.* | 0.02 | 0.05 | 0.04 | 0.07 | 0.06 | 0.10 | 0.33 | 0.27 | 0.53 | **0.95** | **0.97** | **0.91** | 0.04 | 0.08 | 0.09 | 0.14 | 0.12 | 0.19 |
| GreeDy-*temp.* | 0.13 | 0.41 | 0.08 | 0.32 | 0.30 | 0.29 | 0.52 | 0.58 | 0.72 | **0.95** | 0.95 | 0.87 | 0.22 | 0.50 | 0.17 | 0.49 | 0.47 | 0.45 |
| GreeDy-*spa-temp.* | **0.19** | **0.44** | 0.12 | 0.37 | 0.29 | 0.37 | 0.64 | 0.60 | 0.76 | 0.93 | 0.91 | 0.85 | 0.31 | 0.53 | 0.23 | 0.54 | 0.46 | 0.54 |
| GreeDy-*evnt-impct* | 0.10 | 0.28 | 0.07 | 0.34 | 0.26 | 0.27 | 0.62 | 0.61 | 0.67 | **0.95** | **0.96** | 0.88 | 0.18 | 0.39 | 0.14 | 0.51 | 0.42 | 0.43 |
| CoDy-*rand.* | 0.10 | 0.30 | 0.12 | 0.34 | 0.30 | 0.41 | 0.63 | 0.59 | 0.82 | 0.91 | 0.92 | 0.82 | 0.19 | 0.43 | 0.24 | 0.52 | 0.47 | 0.58 |
| CoDy-*temp.* | 0.13 | 0.36 | 0.11 | 0.38 | 0.36 | 0.46 | 0.64 | 0.58 | 0.83 | 0.92 | 0.93 | 0.84 | 0.23 | 0.49 | 0.22 | 0.55 | 0.54 | 0.62 |
| CoDy-*spa-temp.* | **0.19** | 0.43 | **0.16** | 0.39 | 0.35 | 0.50 | **0.67** | **0.63** | **0.84** | 0.92 | 0.90 | 0.82 | **0.31** | **0.54** | **0.30** | 0.57 | 0.52 | 0.65 |
| CoDy-*evnt-impct* | 0.16 | 0.38 | 0.14 | **0.40** | **0.39** | **0.52** | 0.65 | 0.61 | 0.82 | 0.92 | 0.90 | 0.85 | 0.27 | 0.50 | 0.27 | **0.58** | **0.57** | **0.68** |

ative scores, suggesting explanations that preserve model predictions, instead of altering them. Additionally, the provided code is incomplete. We thus decided not to include TempME as a baseline.

**Counterfactual Explainers**  We configure GreeDy with a candidate event limit of 64, sampling up to 10 events per iteration. For CoDy, we also limit the search space to 64 events, with a maximum of 300 iterations and $\alpha = \frac{2}{3}$ to emphasize exploration over exploitation. Appendix G shows a sensitivity analysis on these parameters.

**Explained Instances**  Recognizing that explanations can differ between correct and incorrect predictions (Amara et al., 2022), we separately evaluate instances where the TGNN makes correct versus incorrect predictions. This distinction allows for a more comprehensive assessment of the robustness and reliability of explanation methods across different prediction outcomes.

### 5.2. Evaluation Framework

Factual and counterfactual explanations inherently address different aspects of model predictions. Factual explanations aim to identify the sufficient information for a prediction, while counterfactual explanations focus on identifying the necessary information (Tan et al., 2022). Our evaluation framework thus comprises metrics that allow a nuanced evaluation of necessity and sufficiency. **Sparsity** measures the complexity of an explanation by measuring the proportion of utilized features relative to the total number of available features. Sparsity scores range from 0 to 1, with lower scores indicating more concise explanations. **Fidelity** measures how effectively an explanation identifies key input characteristics that drive predictions. Based on the definitions of the probability of necessity and the probability of sufficiency introduced by Tan et al. (2022), we adopt two fidelity metrics, $fid_-$ and $fid_+$. $fid_-$ measures how well the explanations capture sufficient features required so that the predictions remain unchanged (Amara et al., 2022).

In contrast, $fid_+$ assesses if removing the identified features changes the model's prediction, thus capturing the necessity (Amara et al., 2022). We assess the relationship between fidelity and sparsity by calculating the **Area Under the Fidelity-Sparsity Curve (AUFSC)** as $AUFSC_+$ and $AUFSC_-$, respectively. We also report the **Characterization Score** *char*, which is the harmonic mean of $fid_-$ and $fid_+$ (Amara et al., 2022), and provides an integrated assessment of sufficiency and necessity.

Formal definitions of the metrics are provided in Appendix D. Appendix F provides additional evaluations on explainer runtime, the impact of search iterations, and the similarities between explanations from different explainers. Positive results further highlight CoDy's strong performance.

### 5.3. Results

#### 5.3.1. NECESSITY OF EXPLANATIONS

Table 1 presents a comprehensive overview of the $AUFSC_+$ scores in various experimental scenarios. CoDy generally achieves the highest $AUFSC_+$ scores, except for correct predictions in the UCI-Forums dataset, where GreeDy-*spatio-temporal* (0.44) narrowly exceeds the result of CoDy-*spatio-temporal* (0.43). This hightlights the proficiency of CoDy in providing concise necessary explanations. We observe a consistent trend of CoDy-*spatio-temporal* excelling in explaining correct predictions, while CoDy-*event-impact* performs better in elucidating incorrect predictions, albeit by minor margins (0.01–0.04). This hints at the higher importance of events that are spatially and temporally close to a correctly predicted future link, whereas for incorrect predictions, this seems to be of slightly less importance. For GreeDy, the *spatio-temporal* selection policy generally yields the best results. In general, factual baselines show poor performance in $AUFSC_+$ scores; T-GNNExplainer, despite being specifically developed for TGNNs, averages 75.05% lower than the top-performing explainer, CoDy-*spatio-temporal*.

The disparity between explanations for correct and incor-

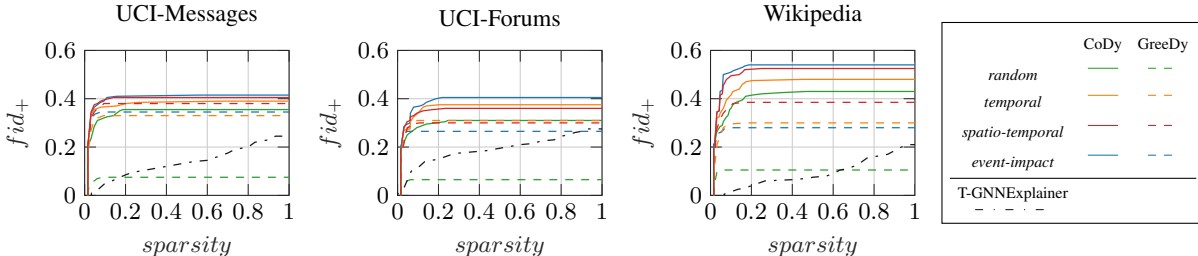

Figure 3. Cumulative $fid_+$ score relative to an upper $sparsity$ limit for incorrect predictions with TGN as target model. PGExplainer is excluded since it is assessed with a fixed sparsity.

rect predictions is particularly pronounced in UCI-Messages and Wikipedia. All explainers identify necessary explanations for incorrect predictions more than twice as often as for correct ones. This suggests a fundamental difference between correct and incorrect predictions. For incorrect predictions, the model tends to misinterpret past information, making explanations easier to identify. In contrast, identifying a counterfactual example for correct predictions requires omitting information to render the prediction incorrect, which can be challenging if most past data align with the prediction. Despite this, explaining incorrect predictions is often more insightful, as it reveals misleading input features and uncovers model limitations. This understanding can inform model improvements and enhance robustness by highlighting areas where the model is vulnerable. Additionally, focusing on incorrect predictions allows for a clearer examination of the decision-making process, revealing potential biases and providing actionable insights for refining the model.

The superior performance of CoDy is further illustrated in Figure 3, which shows the $fid_+$ score achieved with explanations up to a given sparsity level. At sparsity 1, all necessary explanations are considered. All CoDy variants, especially *spatio-temporal* and *event-impact*, exhibit a rapid increase in $fid_+$ at low sparsity levels, highlighting CoDy's effectiveness in delivering concise explanations. T-GNNExplainer fails to achieve comparable performance, even at high sparsity levels, as it often neglects the most critical features relevant to the model's predictions.

Overall, CoDy, particularly with the *spatio-temporal* and *event-impact* policies, excels at balancing the dual objectives of maximizing counterfactual example discovery and minimizing explanation complexity. Unlike GreeDy, which rigidly follows its initial search direction, CoDy dynamically adapts to information gathered during search. Although GreeDy may occasionally outperform CoDy when its early path aligns with the optimal solution, CoDy's flexibility generally leads to better results. Further experiments show that CoDy's performance can be further improved by increasing search iterations (see Appendix F.3) and hyperparameter tuning (see Appendix G).

### 5.3.2. SUFFICIENCY OF EXPLANATIONS

Analyzing the sufficiency of the explanations through the $AUFSC_-$ score, as shown in Table 1, reveals that CoDy consistently achieves high scores of over $0.82$ for incorrect predictions and $0.58$ for correct predictions, indicating that the generated explanations fulfill the sufficiency criteria to a high degree. Among CoDy variants, performance remains notably stable across different settings, with only minor differences observed.

Although factual methods are expected to achieve better $AUFSC_-$ scores, as they aim to identify sufficient explanations for predictions (Tan et al., 2022), PGExplainer and T-GNNExplainer do not consistently outperform their counterfactual counterparts. While T-GNNExplainer achieves impressive $fid_-$ scores of $0.82$ and $0.9$ for correct predictions in UCI-Messages and Wikipedia, it generally falls behind CoDy and GreeDy in $AUFSC_-$. This inconsistency may be attributed to specific implementation details in T-GNNExplainer, particularly an approximation used in model calls, as noted by its original authors (Xia et al., 2023). Such approximations may hinder its ability to fully capture sufficient explanations.

Overall, the strong performance of CoDy and GreeDy underscores their effectiveness in providing concise yet impactful explanations, demonstrating that counterfactual methods excel not only in necessity but also in sufficiency.

### 5.3.3. CONVERGENT ANALYSIS

The characterization score $char$ synthesizes the dimensions of sufficiency and necessity into a single score. Table 1 presents the explainers' performance along this metric. Notably, CoDy-*event-impact* excels in explaining incorrect predictions, whereas CoDy-*spatio-temporal* is superior for correct predictions. The $AUFSC_+$ and $AUFSC_-$ scores reveal that CoDy-*event-impact* and CoDy-*spatio-temporal* not only deliver excellent explanations in terms of necessity but also maintain comparably strong sufficiency. This underscores the significance of the fallback strategy, ensuring the provision of explanations even in the absence of counterfactual examples. These fallback explanations still deliver

mostly sufficient explanations. Thus, even though not all explanations are counterfactual, they consistently highlight pertinent input information.

### 5.3.4. COMPARISON OF SELECTION POLICIES

Selection policies significantly impact the performance of GreeDy and CoDy. The *random* policy yields the poorest results for either method. For GreeDy, applying the *temporal* selection policy improves the $AUFSC_+$ score by $386\%$ on average, while the *spatio-temporal* and *event-impact* policies yield improvements of $485\%$ and $304\%$, respectively, over the *random* policy. CoDy exhibits a similar trend, albeit with smaller average improvements: $14.3\%$ for *temporal*; $36.7\%$ for *spatio-temporal*; $29.6\%$ for *event-impact*. This underscores the importance of spatial and temporal information for explanations. Across all tasks, CoDy-*spatio-temporal* outperforms the strongest baseline (i.e., GreeDy-*spatio-temporal*) by $16\%$ in $AUFSC_+$.

The results demonstrate that CoDy robustly adapts its search based on initial results, exploring different paths if necessary. GreeDy adheres to its initial path, making it more reliant on the selection policy and susceptible to local optima.

Overall, the evaluation shows that CoDy consistently outperforms baseline methods, validating its effectiveness in generating factual and counterfactual explanations. This approach not only improves the identification of necessary and sufficient information but also shows the importance of leveraging spatio-temporal and event-impact insights to effectively navigate the complex search space.

### 5.3.5. PRACTICAL CONSIDERATIONS

In practice, CoDy emerges as the most robust and reliable choice. While it may require more computation when searching for minimal sparsity, its performance remains consistent across diverse scenarios. A practical implementation can significantly reduce runtime by terminating the search upon finding the first valid counterfactual, while still retaining CoDy's strong explanatory capabilities.

If computational efficiency and rapid explanation generation are the primary concerns, GreeDy offers a compelling, faster alternative. It can yield good results quickly, especially when its initial greedy choices align with an effective counterfactual path. However, this speed comes at the cost of a higher risk of suboptimal explanations and a greater likelihood of getting stuck in local optima.

The *spatio-temporal* selection strategy is the most suitable choice for real-world applications where ground-truth labels are not available. It performs best on correct predictions-which are most common in properly functioning TGNNs-and only slightly underperforms the *event-impact* policy when explaining incorrect predictions.

## 6. Conclusion

In this paper, we introduced CoDy, a counterfactual explanation method tailored for Temporal Graph Neural Networks (TGNNs) operating on Continuous-Time Dynamic Graphs. CoDy adapts Monte Carlo Tree Search to generate concise and actionable explanations. Our extensive evaluation demonstrates that CoDy outperforms existing factual explanation methods, such as PGExplainer and T-GNNExplainer, as well as GreeDy, a novel counterfactual baseline based on greedy search. We further show that incorporating spatio-temporal and event-impact information effectively guides the search process, significantly enhancing the quality of counterfactual explanations for TGNNs.

While our evaluation confirms CoDy's ability to identify compact counterfactual examples, future work could investigate its practical utility through user studies in real-world applications. Additionally, although CoDy currently focuses on generating explanations by removing graph components, it could be extended to also support operations such as adding nodes or edges, modifying features, or adjusting event timestamps. These extensions would substantially enlarge the search space, necessitating the development of new strategies to manage the increased complexity.

Despite these challenges, CoDy and its selection policies offer strong potential to advance the interpretability of TGNNs. By providing deeper insights into model decisions, CoDy can support the adoption of TGNNs in high-stakes domains such as finance, healthcare, and scientific research.

## Impact Statement

This work contributes to the broader goal of interpretable machine learning by introducing a novel counterfactual explanation method for temporal graph neural networks (TGNNs). As TGNNs are increasingly applied in high-stakes domains, such as finance and healthcare. CoDy provides actionable, model-agnostic explanations that can help users trust, debug, and audit temporal models. This work promotes transparency in evolving decision environments and may serve as a foundation for future human-in-the-loop or regulatory-compliant AI systems.

## Acknowledgements

The authors acknowledge support from the state of Baden-Württemberg through bwHPC and conducted substantial parts of this work at the Karlsruhe Institute of Technology. This work was funded by the Federal Ministry of Education and Research of Germany and the Saxon State Ministry for Science, Culture, and Tourism as part of the Center for Scalable Data Analytics and Artificial Intelligence Dresden/Leipzig (ScaDS.AI, project ID: SCADS24B).

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

## A. Details of the GreeDy Algorithm

---

**Algorithm 2** GreeDy search algorithm for counterfactual examples.

---

**Input:** TGNN model $f$, input graph $\mathcal{G}$, explained event $\varepsilon_i$, selection policy $\delta$, number of events to sample in each iteration $l$
**Output:** best explanation found $\mathcal{X}$
$p_{orig} \leftarrow f(\mathcal{G}(t_i), \varepsilon_i)$  $n_{root} \leftarrow (\varnothing, p_{orig}, null, \varnothing)$
$n_{best} \leftarrow n_{root}$;
**while** $s_{best}$ does not include all candidate events $C(\mathcal{G}, \varepsilon_i, k, m_{max})$ **do**
    $children_{best} \leftarrow$ set of $l$ child nodes with highest rank according to $\delta$, each child $n_{child}$ initialized with prediction
    $p_{child} = f((\mathcal{G}(t_i) \setminus s_{child}), \varepsilon_i)$  $n_{best\_child} \leftarrow \arg\max_{n_j \in children_{best}} \Delta(p_{orig}, p_j)$  **if** $\Delta(p_{best}, p_{best\_child}) > 0$ **then**
        `/*` $n_{best\_child}$ `shifts the prediction further towards the opposite sign of the original`
          `prediction` `*/`
        $n_{best} \leftarrow n_{best\_child}$ **if** $\Delta(p_{orig}, p_{best}) > |p_{orig}|$ **then**
          `/*` $s_{best}$ `is a counterfactual example` `*/`
        **break**
    **end**
    **else**
        **break**
    **end**
**end**
**return** $s_{best}$

---

Figure 4 depicts the operation of GreeDy. Each iteration involves selecting three past events ($l = 3$) using a policy $\delta$. Starting from the root node, node $n_2$ is chosen as the best node ($n_{best}$) in the first iteration, with a perturbation set $s_2 = \varepsilon_2$ and a prediction score of 1.996. As iterations proceed, the prediction score of $n_{best}$ decreases until it becomes negative in the third iteration. At this point, the search ends, yielding the counterfactual example $\mathcal{X} = s_9 = \varepsilon_2, \varepsilon_1, \varepsilon_5$.

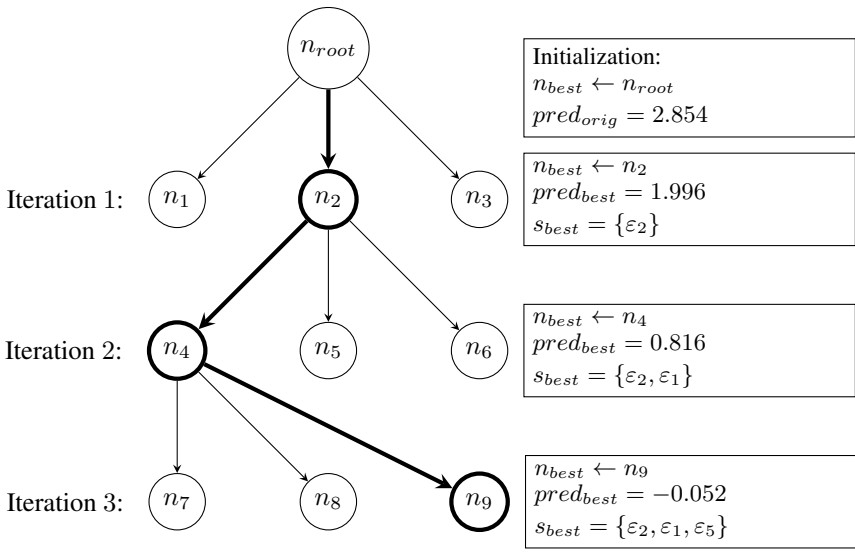

*Figure 4.* Example for the operation of the GreeDy approach.

## B. Details of the CoDy Algorithm

CoDy models nodes in the search tree as tuples. A node $n_j$ is defined by the tuple:

$$n_j = (s_j, p_j, parent_j, children_j, selections_j, score_j, selectable_j) \tag{10}$$

$s_j$ denotes the perturbation set (i.e., set of past events) associated with $n_j$. $parent_j$ denotes the parent node in the search tree, while $children_j$ denotes the set of child nodes in the search tree. $selections_j$ is an integer value that tracks how often the specific node has been selected. $selectable_j$ indicates whether the node is still considered for selection (for example, it does not make sense to select a node that already constitutes a counterfactual example, since its child nodes can only contain counterfactual examples with higher complexity).

Algorithm 3 shows the recursive child node selection process. Child nodes are first ranked by their $selection_{score}$ and ties in this score are broken using a selection policy $\delta$.

---

**Algorithm 3** Recursive selection algorithm.

---

**Function** *select*($n_j$, $\delta$) :
   **if** $n_j$ *is not yet expanded* **then**
     |  return $n_j$;
   **end**
   $n_{best} \leftarrow \arg\max_{n_k \in \ selectable\_children(n_j)} sel\_score(n_k)$  **if** *there is more than one child with the highest selection score* **then**
     |  $n_{best} \leftarrow$ highest ranking child node according to selection strategy $\delta$
   **end**
   return **select**($n_{best}$)
**end**

---

The simulation algorithm (Algorithm 4) consists of calling the explained TGNN to predict the score for the future link given the input graph without the events $s_j$ associated with the selected node $n_j$.

---

**Algorithm 4** Algorithm for simulating the link prediction on the selected node.

---

**Function** *simulate*($n_j$, $f$, $\mathcal{G}$, $\varepsilon_i$) :
  |  $p_j \leftarrow f((\mathcal{G}(t_i) \setminus s_j), \varepsilon_i)$  return $p_j$
**end**

---

Algorithm 5 showcases the expansion process. First, values for $selections_j$ and $score_j$ are set for the selected node $n_j$. Second, depending on whether $s_j$ constitutes a counterfactual example the node is added to the list of counterfactual examples, or expanded further. This expansion entails adding all possible child nodes with initial $null$ values.

---

**Algorithm 5** Function for expanding the selected node.

---

**Function** *expand*($n_j$, $p_{orig}$) :
   $selections_j \leftarrow 1$  $score_j \leftarrow \max\left(0, \frac{\Delta(p_{orig}, p_j)}{|p_{orig}|}\right)$
   **if** $score_j > 1$ **then**
     |  $selectable_j \leftarrow 0$  Add $n_j$ to a list of counterfactual examples $cf\_examples$
   **else**
     |  $children_j \leftarrow \{(s_k, null, n_j, \varnothing, 0, null, 1) : s_k \in \hat{S}_{\varepsilon_j}$ with $|s_k| = |s_j| + 1, s_j \subset s_k\}$  **if** $children_j = \varnothing$ **then**
       |  $selectable_j \leftarrow 0$
     **end**
   **end**
**end**

---

The backpropagartion algorithm (Algorithm 6) recursively updates the $score_j$, $selections_j$, and $selectable_j$ values of each traversed node $n_j$.

---

**Algorithm 6** Backpropagation function that recursively updates the information of nodes in the search tree.

---

**Function** *backpropagate*$(n_j)$ :

    **if** $n_j = null$ **then**

        | **return**

    **end**

    $selections_j \leftarrow selections_j + 1$   $score_j \leftarrow \max\left(0, \frac{\Delta(p_{orig}, p_j)}{|p_{orig}|}\right)$   $score_j \leftarrow score_j + \sum_{n_k \in children_j}(score_k *$

    $selections_k)$  $score_j \leftarrow \frac{score_j}{selections_j}$   **if** *No child in $children_j$ is selectable* **then**

        |   $selectable_j \leftarrow 0$

    **end**

**end**

---

## C. Dataset Statistics

The explainers are evaluated on three different datasets. These datasets are diverse in terms of their size, their structure, and the temporal density of events, aiming to verify that the explanation approaches perform similarly on different datasets.

Table 2 provides an overview of the key statistics of datasets. The two datasets from the UCI social network cover a longer timespan and are more sparse in the temporal dimension than the Wikipedia dataset. Another substantial difference is that, in contrast to the other datasets, the UCI-Messages dataset contains a relatively high number of unique edges compared to the number of total edges. Furthermore, the UCI-Messages and UCI-Forums datasets have proportionally fewer multi-edges compared to the Wikipedia dataset.

*Table 2.* Statistics for datasets

| Dataset | # Nodes | # Edges | # Unique Edges | Timespan | Graph Type |
|---|---|---|---|---|---|
| UCI-Messages (Kunegis, 2013) | 1,899 | 59,835 | 20,296 | 196 days | Unipartite |
| UCI-Forums (Kunegis, 2013) | 1,421 | 33,720 | 7,089 | 165 days | Bipartite |
| Wikipedia (Kumar et al., 2019) | 9,227 | 157,474 | 18,257 | 30 days | Bipartite |

# D. Evaluation Framework

We evaluate the explanation methods using $sparisty$, $fid_+$, $fid_-$, $AUFSC_+$, $AUFSC_-$, and $char$.

**Sparsity:**   Sparsity is a widely used metric to gauge the complexity of explanations (Prado-Romero et al., 2024; Yuan et al., 2023; Amara et al., 2022), capturing how well the complexity of explanations is minimized. We define it as the mean ratio between the size of explanations $|\mathcal{X}_{\varepsilon_i}|$ and the size of the full set of candidate events $|C(\mathcal{G}, \varepsilon_i, k, m_{max})|$:

$$sparsity = \frac{1}{N} \sum_{i=1}^{N} \frac{|\mathcal{X}_{\varepsilon_i}|}{|C(\mathcal{G}, \varepsilon_i, k, m_{max})|} \tag{11}$$

**Fidelity:**   The fidelity metrics capture how well the explanations capture pertinent events. Since we are comparing factual and counterfactual explainers we adapt the definions for the probability of sufficiency and the probability of necessity introduced by Tan et al. (2022) and map them to two fidelity metrics, $fid_+$ and $fid_-$, for the context of temporal graphs. In line with the definitions of Amara et al. (2022) we assess fidelity with regards to the predictions of the model, not in regards to the ground truth labels. This is sometimes termed "correctness" or "validity" Prado-Romero et al. (2024).

$$fid_- = \frac{1}{N} \sum_{i=1}^{N} \mathbb{1}(p(f(\mathcal{G}(t_i)), \varepsilon_i) = p(f(\mathcal{X}_{\varepsilon_i}, \varepsilon_i))) \tag{12}$$

$$fid_+ = 1 - \frac{1}{N} \sum_{i=1}^{N} \mathbb{1}(p(f(\mathcal{G}(t_i)), \varepsilon_i) = p(f((\mathcal{G}(t_i) \setminus \mathcal{X}_{\varepsilon_i}), \varepsilon_i))) \tag{13}$$

In these equations, $\varepsilon_1, ..., \varepsilon_N$ represent the explained future links in an experiment, and $\mathcal{X}_{\varepsilon_1}, ..., \mathcal{X}_{\varepsilon_N}$ denote their corresponding explanations. The indicator function $\mathbb{1}(a = b)$ returns 1, if $a$ equals to $b$, otherwise, it returns 0. Based on the fidelity scores we calculate $AUFSC_+$ and $AUFSC_-$ by integrating over the fidelity-sparsity curve.

**Characterization Score:**   To jointly assess necessity and sufficiency, we adopt the characterization score $char$ introduced by Amara et al. (2022):

$$char = \frac{w_+ + w_-}{\frac{w_+}{fid_+} + \frac{w_-}{fid_-}} \tag{14}$$

Here, $w_+$ and $w_-$ are weights for $fid_+$ and $fid_-$ that allow putting more emphasis on either sufficiency or necessity. To make a fair comparison between the counterfactual explainers and the factual baselines, the weights are set to $w_+ = w_- = 0.5$. The characterization score char takes on values between 0 and 1, where larger values indicate better performance.

# E. Evaluation of Explanation Methods Applied to TGAT

When explaining the TGAT model (Xu et al., 2020) we find similar results to the findings on TGN. The results presented in Table 3 show the counterfactual explanation methods GreeDy and CoDy outperforming the factual baseline T-GNNExplainer regarding sparsity, $fid_+$ and the $char$ score. T-GNNExplainer faires better regarding the $fid_-$ metric, achieving the highest score for correct predictions in the UCI-Messages (0.76) and the Wikipedia (0.87) dataset. In the other settings, the counterfactual explainers achieve better results.

The main difference in the performance of GreeDy and CoDy is that with the evaluated settings, GreeDy-*spatio-temporal* nearly outperforms all CoDy variants regarding $fid_+$ on the UCI-Messages dataset. Looking at the relationship between $fid_+$ and sparsity in Figure 5 shows the reason for the limited performance of CoDy in these settings: While GreeDy and CoDy variants find a similar amount of necessary explanations with low sparsity, GreeDy-*spatio-temporal* also finds more counterfactual examples with higher sparsity. The fact that CoDy does not elucidate explanations with larger sparsities to a similar degree suggests that the explainer is configured sub-optimally. We investigate this further in the sensitivity analysis performed in G and show that changing the balance between exploration and exploitation through parameter $\alpha$, and increasing the candidate size $m_{max}$ can boost the performance of CoDy without increasing search iterations.

*Table 3.* Results on the *fidelity+*, *sparsity*, *fidelity-*, and *char* scores of the different explanation methods for explaining correct and incorrect predictions for the TGAT target model. The best result for each experimental setting is **bold**, and the second best is underlined.

| | Fidelity+ | | | | | | Sparsity | | | | | |
| | UCI-Messages | | UCI-Forums | | Wikipedia | | UCI-Messages | | UCI-Forums | | Wikipedia | |
| | corr. | wrg. | corr. | wrg. | corr. | wrg. | corr. | wrg. | corr. | wrg. | corr. | wrg. |
|---|---|---|---|---|---|---|---|---|---|---|---|---|
| T-GNNExplainer | 0.14 | 0.08 | 0.11 | 0.19 | 0.08 | 0.24 | 0.36 | 0.29 | 0.03 | 0.26 | 0.25 | 0.39 |
| GreeDy-*rand.* | 0.06 | 0.08 | 0.21 | 0.18 | 0.05 | 0.11 | 0.1 | 0.04 | 0.07 | 0.08 | 0.04 | 0.02 |
| GreeDy-*temp.* | 0.17 | 0.26 | 0.42 | 0.36 | 0.10 | 0.30 | 0.07 | 0.04 | 0.05 | 0.05 | 0.06 | 0.03 |
| GreeDy-*spa-temp.* | **0.38** | **0.34** | 0.47 | 0.35 | 0.14 | 0.39 | 0.11 | 0.05 | 0.06 | 0.05 | 0.07 | 0.04 |
| GreeDy-*event-impact* | 0.22 | 0.29 | 0.31 | 0.33 | 0.08 | 0.28 | 0.11 | 0.07 | 0.09 | 0.09 | 0.05 | 0.02 |
| CoDy-*rand.* | 0.17 | 0.26 | 0.38 | 0.34 | 0.15 | 0.35 | 0.03 | 0.04 | 0.22 | 0.06 | 0.08 | 0.05 |
| CoDy-*temp.* | 0.19 | 0.30 | 0.46 | **0.37** | **0.18** | 0.35 | 0.03 | 0.04 | 0.13 | 0.05 | 0.08 | 0.04 |
| CoDy-*spa-temp.* | 0.32 | 0.33 | **0.56** | 0.35 | **0.18** | 0.35 | 0.05 | 0.04 | 0.08 | 0.04 | 0.08 | 0.04 |
| CoDy-*event-impact* | 0.24 | 0.32 | 0.48 | 0.34 | **0.18** | **0.41** | 0.04 | 0.05 | 0.21 | 0.06 | 0.08 | 0.05 |

| | Fidelity- | | | | | | Char | | | | | |
| | UCI-Messages | | UCI-Forums | | Wikipedia | | UCI-Messages | | UCI-Forums | | Wikipedia | |
| | corr. | inc. | corr. | inc. | corr. | inc. | corr. | inc. | corr. | inc. | corr. | inc. |
|---|---|---|---|---|---|---|---|---|---|---|---|---|
| T-GNNExplainer | **0.76** | 0.91 | 0.46 | 0.50 | **0.87** | 0.84 | 0.24 | 0.14 | 0.17 | 0.03 | 0.14 | 0.38 |
| GreeDy-*rand.* | 0.39 | **1.00** | 0.49 | 0.97 | 0.57 | 0.93 | 0.10 | 0.15 | 0.29 | 0.30 | 0.09 | 0.19 |
| GreeDy-*temp.* | 0.57 | **1.00** | 0.67 | **0.99** | 0.77 | 0.90 | 0.26 | 0.41 | 0.51 | 0.53 | 0.17 | 0.45 |
| GreeDy-*spa-temp.* | 0.32 | **1.00** | **0.69** | 0.95 | 0.82 | 0.88 | **0.48** | **0.50** | 0.56 | 0.51 | 0.23 | 0.54 |
| GreeDy-*event-impact* | 0.66 | **1.00** | **0.69** | 0.98 | 0.72 | 0.90 | 0.32 | 0.44 | 0.43 | 0.49 | 0.14 | 0.43 |
| CoDy-*rand.* | 0.66 | **1.00** | 0.76 | 0.96 | 0.81 | **0.97** | 0.26 | 0.41 | 0.51 | 0.50 | 0.25 | 0.51 |
| CoDy-*temp.* | 0.65 | **1.00** | 0.76 | 0.98 | 0.83 | **0.97** | 0.29 | 0.46 | 0.57 | **0.54** | 0.29 | 0.51 |
| CoDy-*spa-temp.* | 0.61 | **1.00** | **0.77** | 0.95 | 0.83 | 0.96 | 0.42 | **0.50** | **0.65** | 0.51 | 0.29 | 0.51 |
| CoDy-*event-impact* | 0.66 | **1.00** | **0.77** | 0.97 | 0.83 | **0.97** | 0.35 | 0.48 | 0.59 | 0.50 | **0.30** | **0.58** |

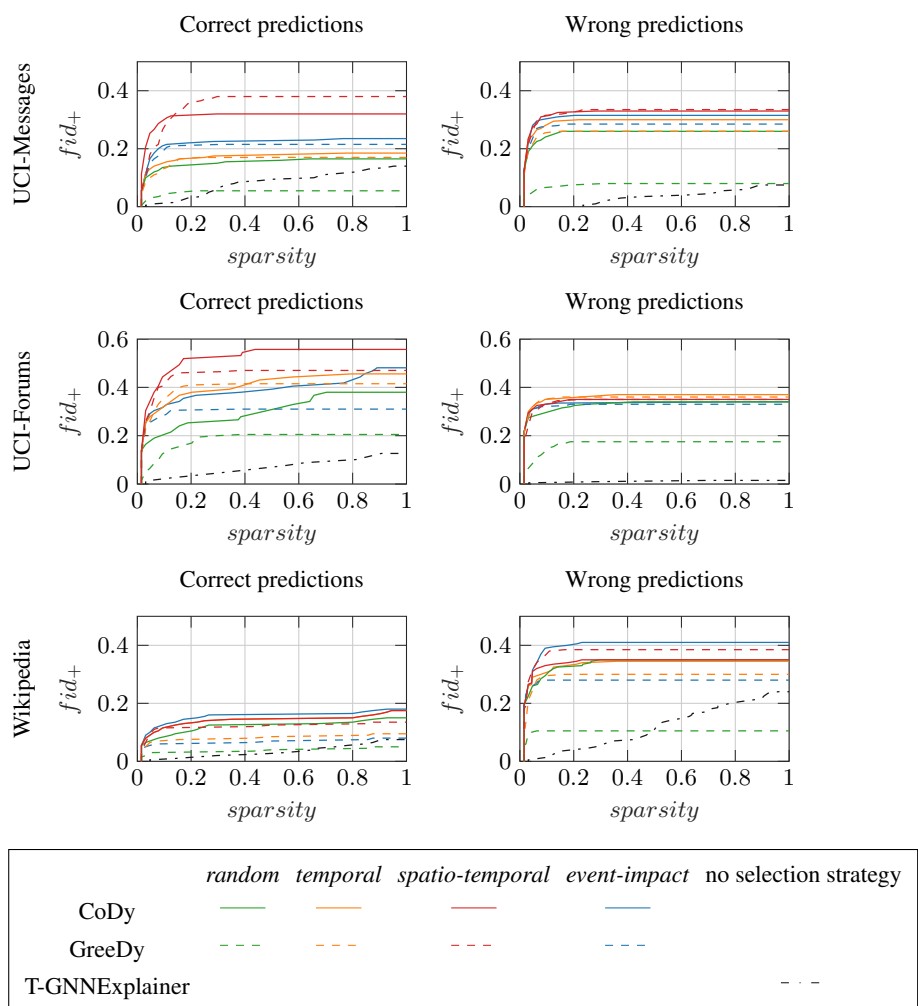

*Figure 5.* Results on the relationship between $fid_+$ and $sparsity$ under different experimental settings for the TGAT model.

# F. Extended Evaluation of Explanation Methods Applied to TGN

We supplement our analysis of the evaluation of our explanation approaches with further results extracted from explaining predictions by the TGN model. Detailed results on the fidelity and sparsity scores are depicted in Table 4. The results on fidelity largely overlap with the results in *AUFSC*.

Figure 7 shows the relationship between $fid_+$ and $sparsity$ for correct predictions. On the UCI-Forums dataset, we see the only setting where a GreeDy variant outperforms all CoDy variants. In F.3, we show that increasing the number of search iterations leads to CoDy outperforming GreeDy in this setting as well.

*Table 4.* Results on the *fidelity+*, *sparsity*, *fidelity-*, and *char* scores of the different explanation methods for explaining correct and incorrect predictions. The best result for each experimental setting is **bold**, and the second best is underlined.

| | Fidelity+ | | | | | | Sparsity | | | | | |
| --- | --- | --- | --- | --- | --- | --- | --- | --- | --- | --- | --- | --- |
| | UCI-Messages | | UCI-Forums | | Wikipedia | | UCI-Messages | | UCI-Forums | | Wikipedia | |
| | corr. | wrg. | corr. | wrg. | corr. | wrg. | corr. | wrg. | corr. | wrg. | corr. | wrg. |
| PGExplainer (Luo et al., 2020) | 0.03 | 0.10 | 0.04 | 0.05 | 0.05 | 0.13 | 0.20 | 0.20 | 0.20 | 0.20 | 0.20 | 0.20 |
| T-GNNExplainer (Xia et al., 2023) | 0.10 | 0.25 | 0.05 | 0.28 | 0.05 | 0.22 | 0.43 | 0.35 | 0.29 | 0.35 | 0.33 | 0.43 |
| GreeDy-*random* | 0.02 | 0.08 | 0.05 | 0.07 | 0.05 | 0.11 | 0.03 | 0.02 | 0.02 | 0.02 | 0.04 | 0.02 |
| GreeDy-*temporal* | 0.14 | 0.33 | 0.43 | 0.31 | 0.10 | 0.30 | 0.05 | 0.04 | 0.05 | 0.03 | 0.06 | 0.03 |
| GreeDy-*spatio-temporal* | **0.20** | 0.38 | **0.46** | 0.30 | 0.14 | 0.39 | 0.08 | 0.04 | 0.05 | 0.04 | 0.07 | 0.04 |
| GreeDy-*event-impact* | 0.11 | 0.35 | 0.29 | 0.27 | 0.08 | 0.28 | 0.06 | 0.03 | 0.04 | 0.03 | 0.05 | 0.02 |
| CoDy-*random* | 0.11 | 0.36 | 0.32 | 0.31 | 0.14 | 0.43 | 0.08 | 0.05 | 0.09 | 0.06 | 0.07 | 0.07 |
| CoDy-*temporal* | 0.14 | 0.39 | 0.39 | 0.38 | 0.13 | 0.48 | 0.07 | 0.04 | 0.10 | 0.05 | 0.06 | 0.06 |
| CoDy-*spatio-temporal* | **0.20** | 0.41 | 0.44 | 0.36 | **0.18** | 0.53 | 0.06 | 0.04 | 0.08 | 0.04 | 0.07 | 0.05 |
| CoDy-*event-impact* | 0.17 | **0.42** | 0.40 | **0.41** | 0.16 | **0.54** | 0.07 | 0.05 | 0.09 | 0.06 | 0.07 | 0.06 |
| | Fidelity- | | | | | | Char | | | | | |
| | UCI-Messages | | UCI-Forums | | Wikipedia | | UCI-Messages | | UCI-Forums | | Wikipedia | |
| | corr. | wrg. | corr. | wrg. | corr. | wrg. | corr. | wrg. | corr. | wrg. | corr. | wrg. |
| PGExplainer (Luo et al., 2020) | 0.49 | 0.76 | 0.44 | 0.84 | 0.87 | 0.68 | 0.05 | 0.17 | 0.07 | 0.08 | 0.09 | 0.22 |
| T-GNNExplainer (Xia et al., 2023) | **0.82** | 0.83 | 0.57 | 0.72 | 0.90 | 0.69 | 0.17 | 0.39 | 0.08 | 0.40 | 0.09 | 0.34 |
| GreeDy-*random* | 0.34 | 0.97 | 0.28 | **0.99** | 0.57 | **0.93** | 0.04 | 0.14 | 0.08 | 0.12 | 0.09 | 0.19 |
| GreeDy-*temporal* | 0.56 | **0.98** | 0.61 | 0.98 | 0.77 | 0.90 | 0.22 | 0.49 | 0.50 | 0.47 | 0.17 | 0.45 |
| GreeDy-*spatio-temporal* | 0.70 | 0.96 | 0.64 | 0.95 | 0.82 | 0.88 | 0.31 | 0.54 | 0.53 | 0.46 | 0.23 | 0.54 |
| GreeDy-*event-impact* | 0.67 | **0.98** | 0.64 | **0.99** | 0.72 | 0.90 | 0.18 | 0.51 | 0.39 | 0.42 | 0.14 | 0.43 |
| CoDy-*random* | 0.68 | 0.95 | 0.66 | 0.97 | 0.88 | 0.88 | 0.19 | 0.52 | 0.43 | 0.47 | 0.24 | 0.58 |
| CoDy-*temporal* | 0.69 | 0.96 | 0.66 | 0.98 | 0.88 | 0.88 | 0.23 | 0.55 | 0.49 | 0.54 | 0.22 | 0.62 |
| CoDy-*spatio-temporal* | 0.72 | 0.95 | **0.69** | 0.93 | **0.91** | 0.86 | **0.31** | 0.57 | **0.54** | 0.52 | **0.30** | 0.65 |
| CoDy-*event-impact* | 0.70 | 0.96 | 0.68 | 0.95 | 0.88 | 0.90 | 0.27 | **0.58** | 0.50 | **0.57** | 0.27 | **0.68** |

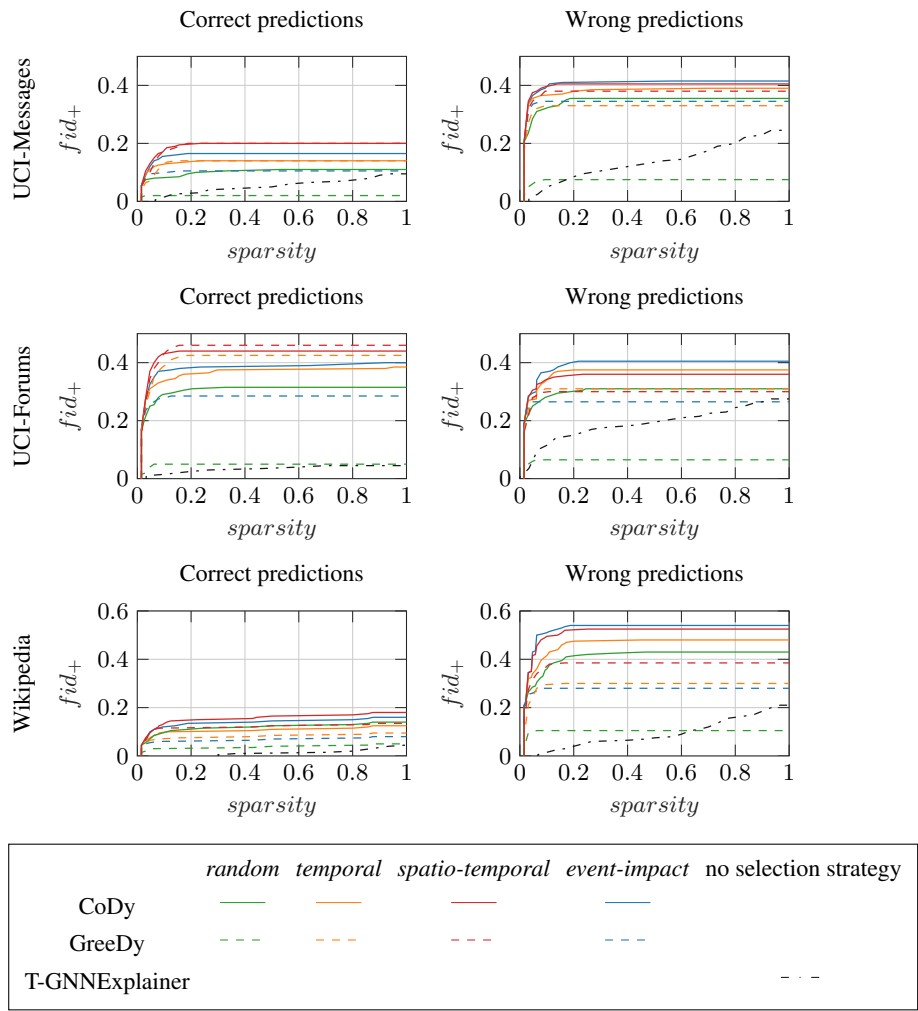

Figure 6. Results on the relationship between $fid_+$ and $sparsity$ under different experimental settings for the TGN model.

## F.1. Fidelity+ to Sparsity for correct predictions

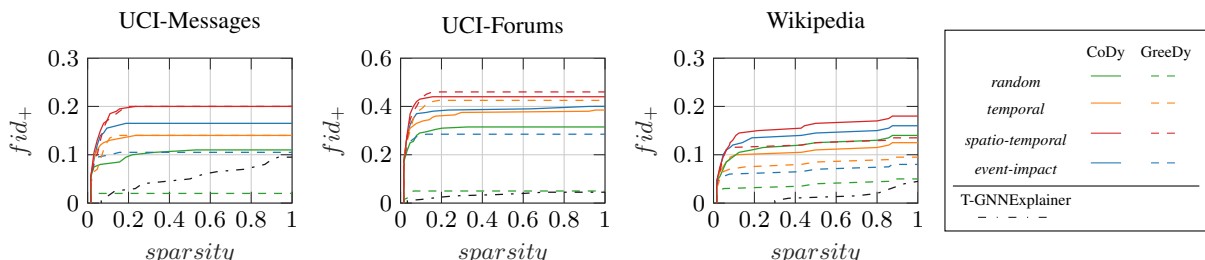

Figure 7. Results on the relationship between $fid_+$ and $sparsity$ for correct predictions made by the TGN model.

## F.2. Runtime

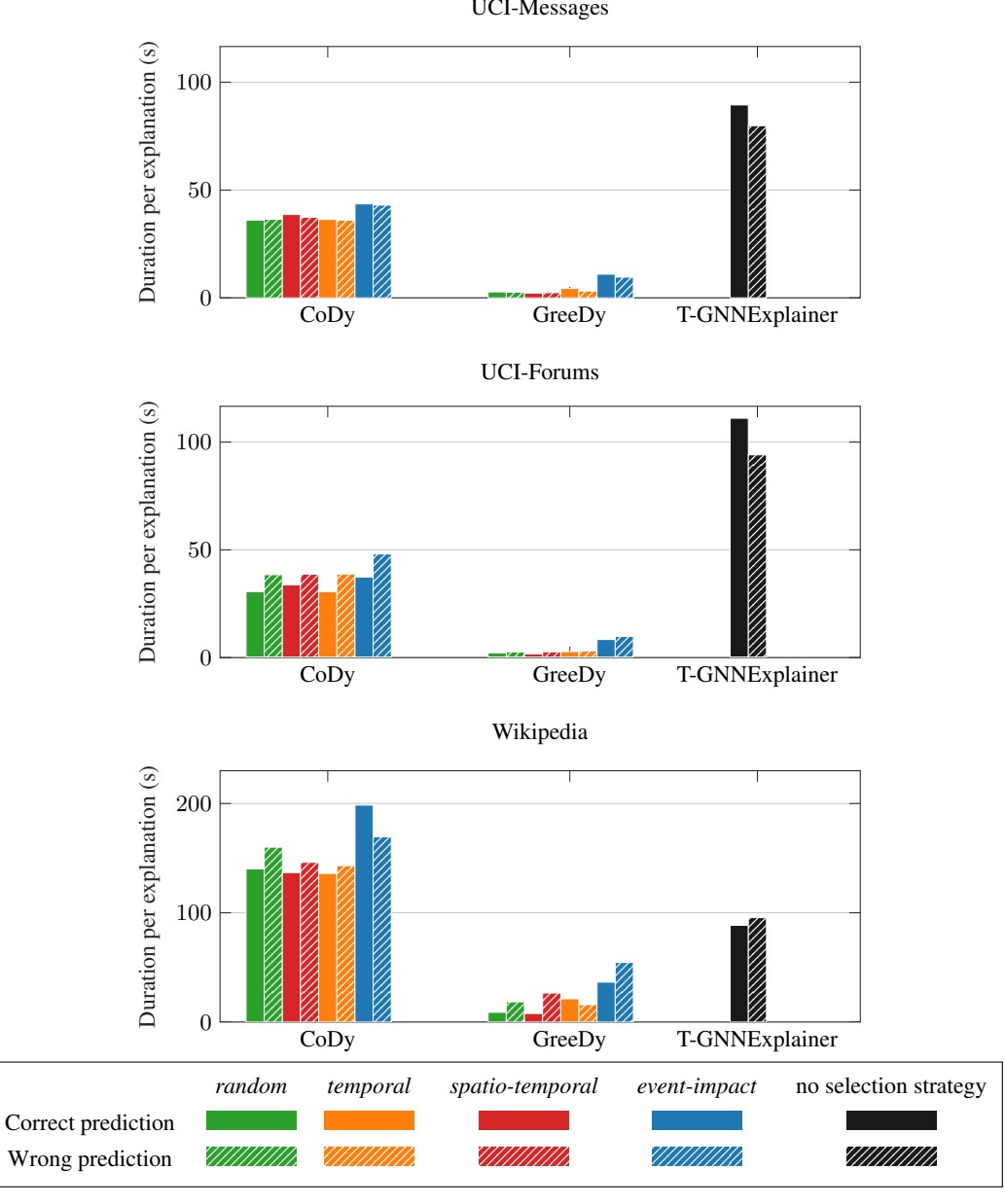

*Figure 8.* Average duration for explaining predictions across various settings when targeting the TGN model.

We report the runtime achieved during the evaluation of the explainers. For replicability, we run the experiments on a high-performance computing cluster with an Intel Xeon Gold 6230 CPU, 16GB of RAM, and an NVIDIA Tesla V100 SXM2 GPU with 32GB of VRAM.

Figure 8 presents the average explanation times across datasets. GreeDy consistently requires the least time, with its *event-impact* variant taking longer due to more frequent calls to the TGNN, as detailed in Table 5. In comparison, CoDy calls this function more often than GreeDy, which concludes explanations upon finding a counterfactual example or reaching a search impasse. CoDy, in contrast, continues until a set number of iterations are completed or the search tree is fully explored.

The potential runtime increase of CoDy over GreeDy is largely a configurable design choice. In our experiments, CoDy continues searching for a better explanation, even after a valid counterfactual has already been found. Thus, the runtime of CoDy could be substantially reduced by only searching until finding the first counterfactual example instead of searching for the predefined number of maximum iterations. Further, a hybrid approach of CoDy and GreeDy could be employed to speed up the search. By first employing GreeDy and, if unsuccessful, running CoDy, starting with the existing partial search tree initialized with GreeDy, one could harness the performance of GreeDy, while benefiting from the better performance of CoDy.

CoDy, GreeDy, and T-GNNExplainer allocate 97.85%, 99.83%, and 82.72% of their time, respectively, to calling the TGNN. This indicates that CoDy's scalability is largely dependent on the underlying TGNN model, which may encounter difficulties with larger or more complex datasets (as seen in the Wikipedia dataset in Figure 8). By efficiently navigating the search space through a balanced approach to exploration and exploitation, along with heuristic selection policies, CoDy optimizes the explanation process, which is especially important for complex and suboptimal TGNN implementations. Notably, T-GNNExplainer dedicates more time to the search process than the other methods.

Since calling the TGNN constitutes the predominant part of the runtime we analyze the time complexity of CoDy in terms of the number of calls to the TGNN. Following Algorithm 1, CoDy has a worst-case time complexity of $O(it_{max})$ in all variants except for the *event-impact* where the initial local gradient exploration of the $N$ candidate event adds to the complexity, resulting in a worst-case complexity of $O(it_{max} + N)$.

Table 5 shows consistent call frequencies to the prediction model across datasets for each explainer. However, CoDy and GreeDy show significant variation in explanation times across datasets, especially on the Wikipedia dataset. This could be due to the dataset's longer model response times, possibly because of its edge features and larger size compared to the other datasets. Note that PGExplainer and T-GNNExplainer are excluded from this table because PGExplainer does not directly interact with TGNNs, and T-GNNExplainer relies on an approximation of the TGNN models.

Overall, GreeDy offers the shortest explanation durations, making it preferable for rapid explanations.

*Table 5.* Average number of calls to the TGNN performed by the explanation methods for the different experimental settings performed by GreeDy and CoDy.

| | UCI-Messages | | UCI-Forums | | Wikipedia | |
|---|---|---|---|---|---|---|
| | Correct | Incorrect | Correct | Incorrect | Correct | Incorrect |
| GreeDy-*random* | 27.10 | 23.40 | 24.80 | 21.80 | 23.15 | 18.55 |
| GreeDy-*temporal* | 43.70 | 29.85 | 36.15 | 29.15 | 32.30 | 24.65 |
| GreeDy-*spatio-temporal* | 57.60 | 29.40 | 37.65 | 35.25 | 39.35 | 22.35 |
| GreeDy-*event-impact* | 100.15 | 79.18 | 88.10 | 78.15 | 80.47 | 69.36 |
| CoDy-*random* | 287.91 | 246.83 | 261.13 | 257.23 | 284.52 | 237.16 |
| CoDy-*temporal* | 287.92 | 245.64 | 261.23 | 256.31 | 284.85 | 236.26 |
| CoDy-*spatio-temporal* | 287.95 | 245.44 | 260.95 | 256.44 | 284.98 | 236.26 |
| CoDy-*event-impact* | 346.50 | 292.23 | 312.00 | 305.70 | 339.90 | 272.52 |

### F.3. Search Iterations

In the experiments with CoDy, a key parameter is the maximum number of search iterations ($it_{max}$), initially set at 300. Further investigation on performance involved increasing $it_{max}$ to 1200, specifically focusing on correct predictions in the Wikipedia dataset where GreeDy-*spatio-temporal* surpasses all CoDy variants. Figure 9 illustrates the $fid_+$ scores relative to the number of iterations, highlighting that CoDy variants eventually outperform GreeDy counterparts, albeit with varying iteration requirements. While *random* and *event-impact* policies require fewer iterations, CoDy-*spatio-temporal* and CoDy-*temporal* need close to 1000 and 1200 iterations, respectively, for superiority.

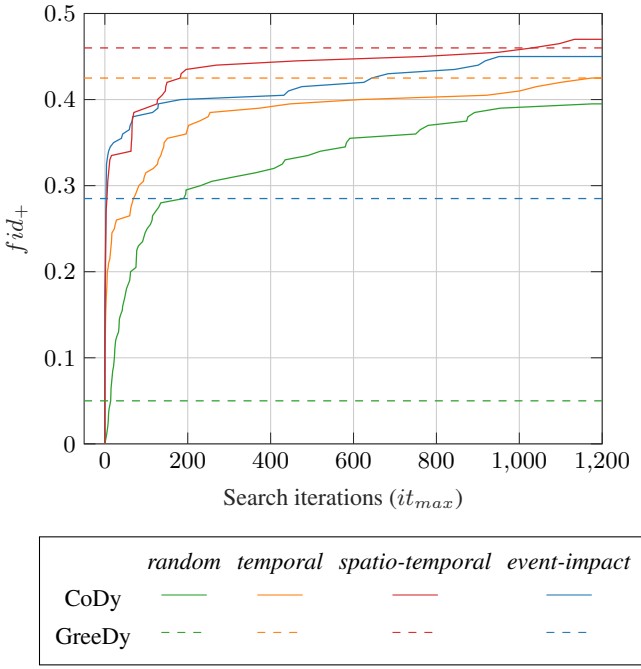

*Figure 9.* Comparison of $fid_+$ scores for various explanation methods on the Wikipedia dataset and the TGN model, highlighting CoDy's performance over iterations.

The figure reveals that CoDy can potentially find the minimal counterfactual example with unlimited iterations, by exploring the entire search space. However, the high iteration count for certain CoDy variants to outperform GreeDy indicates a need for optimizing the parameter $\alpha$, which balance exploration and exploitation. This is inferred from GreeDy's initial superiority, suggesting its more effective exploration of the search space. CoDy's delay in matching GreeDy's partial search tree performance implies an imbalance in its selection policy.

## F.4. Similarities in Explanations

The Jaccard similarity is used to compare explanations from various explainers, focusing on correct predictions in the Wikipedia dataset (Figure 10). This heatmap serves as a representative example. T-GNNExplainer's explanations differ significantly from those of GreeDy and CoDy, partly due to their lower sparsity (Table 1). GreeDy-*temporal* and GreeDy-*spatio-temporal* show high similarity, as do GreeDy-*spatio-temporal* and CoDy explanations.

Comparatively, CoDy variants' explanations are more similar to each other. CoDy-*temporal* and CoDy-*spatio-temporal* have high similarities with each other and with CoDy-*event-impact*. This indicates CoDy's search procedure's ability to explore similar areas, highlighting the likeness of *temporal*, *spatio-temporal*, and *event-impact* policies over the *random* policy.

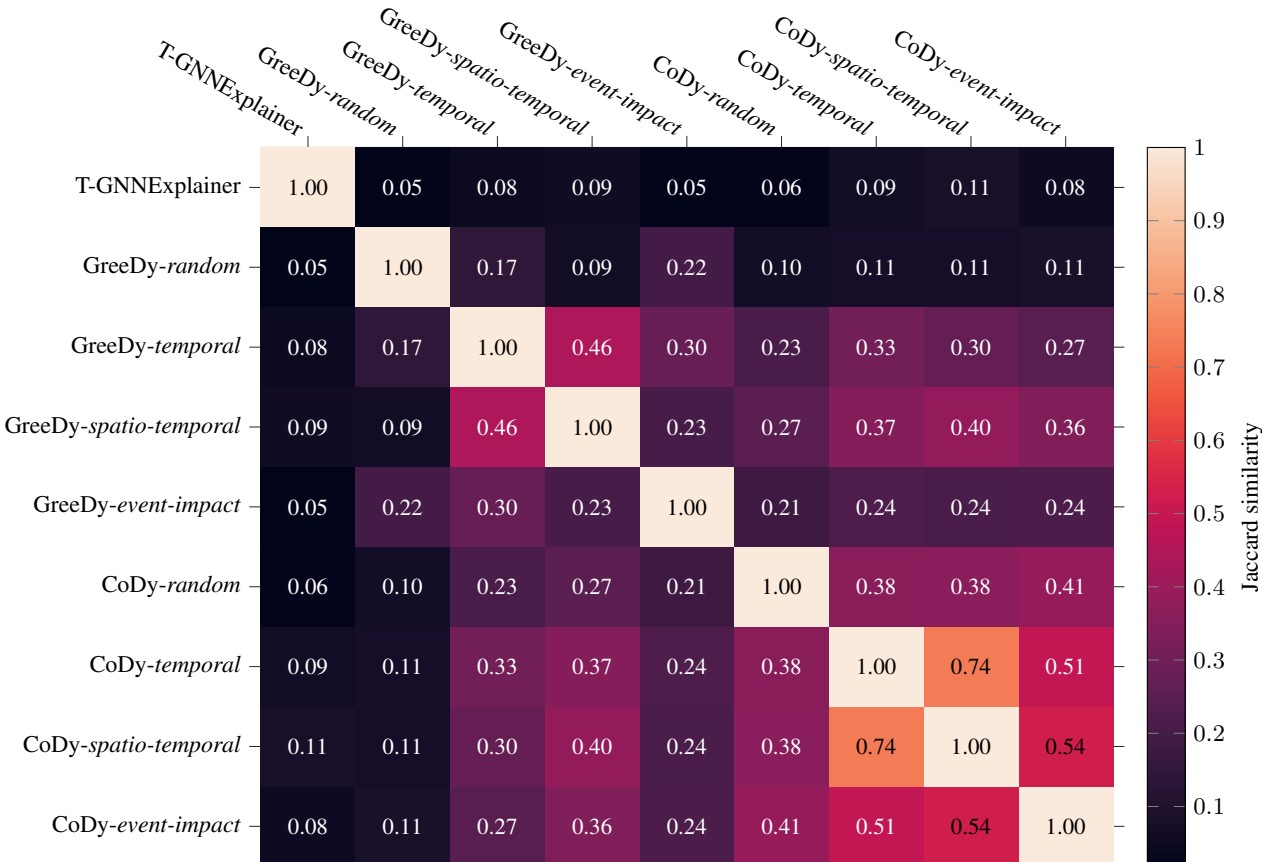

*Figure 10.* Results on the Jaccard similarity between explanations of correct predictions on the Wikipedia dataset for explaining the TGN model.

# G. Sensitivity Analysis

For running the experiments, we did not tune the parameters of CoDy and instead relied on informed guesses for suitable parameters. To gauge the impact of tuning the exploration parameter $\alpha$, as well as the influence of the size of the constraint search space $m_{max}$ in a sensitivity analysis. For this analysis, we ran experiments explaining correct predictions of the TGAT model on the UCI-Messages dataset with the *spatio-temporal* selection policy. The base experiment uses hyperparameters $\alpha = \frac{2}{3}$ and $m_{max} = 64$. To shift the balance between exploration and exploitation we run an experimentation with a stronger emphasis on exploration with $\alpha = \frac{1}{2}$, $m_{max} = 64$, and an experiment with a stronger emphasis on exploitation with $\alpha = \frac{3}{4}$, $m_{max} = 64$. To explore different search-space sizes, we run an experiment with $m_{max} = 48$, and one with $m_{max} = 96$.

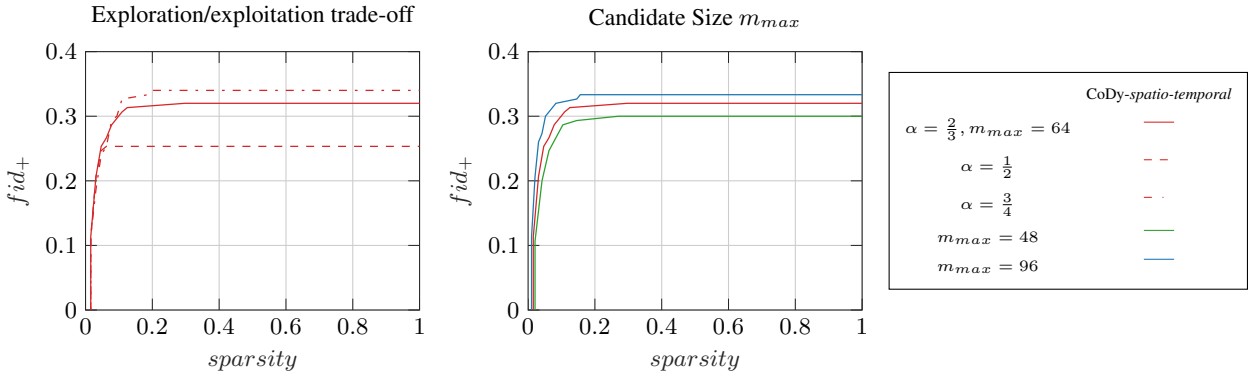

*Figure 11.* $fid_+$-$sparsity$ plots for experiments in sensitivity analysis.

Looking at the influence of adjusting $\alpha$, we see that incentivizing exploitation (larger $\alpha$) improves the performance of CoDy. This suggests that the experimental configuration is suboptimal and misses explanations that require more aggressive exploitation of the partial search tree.

Similarly, the candidate size $m_{max}$ has an influence on the performance of CoDy. Here, increasing the candidate size to 96 improves performance, whereas lowering it to 48 deteriorates the performance. These results hint that there exist explanations to be found beyond the limit of 64 candidates. Increasing the candidate size can thus improve performance, which is the case in the evaluated setting.

Overall, the results of the sensitivity analysis highlight that CoDy has the potential to achieve significantly higher scores by tuning the parameters to the specific data and model. However, as shown in our evaluation, CoDy achieves state-of-the-art performance for counterfactual explanations on dynamic graphs even without such tuning efforts.

