# OpenReview forum: "CoDy: Counterfactual Explainers for Dynamic Graphs"
_ICML.cc/2025/Conference — ICML 2025 poster_

### Official Review · Reviewer_5zq2 · 2025-03-04

**Overall Recommendation:** 3

**Summary:**

This paper introduces CoDy (Counterfactual Explainer for Dynamic Graphs), a method for generating counterfactual explanations to interpret predictions made by Temporal Graph Neural Networks (TGNNs) on continuous-time dynamic graphs (CTDGs). Existing explanation methods focus on static graphs or factual explanations, which are insufficient for capturing temporal dependencies and actionable scenarios in dynamic graphs. To close this gap, CoDy combines Monte Carlo Tree Search (MCTS) with heuristic policies to efficiently explore the search space of past events, identifying minimal subsets of events whose removal alters the model’s prediction. The authors also develop a greedy baseline that iteratively selects events causing the largest immediate prediction change. Experimental results empirically demonstrate the superior performance of CoDy in identifying concise yet impactful counterfactual explanations.

## update after rebuttal
Thanks for the authors' efforts in the rebuttal. I intend to keep my rating.

**Claims And Evidence:**

The main claims made in this paper are generally supported by clear and convincing evidence.

**Essential References Not Discussed:**

Most of the key related work are cited.

**Experimental Designs Or Analyses:**

The soundness of all the experimental designs and analyses has been carefully checked.

**Methods And Evaluation Criteria:**

The proposed method and evaluation criteria are well-aligned with the problem of explaining TGNN predictions on dynamic graphs.

**Other Comments Or Suggestions:**

The notation p in Equation (6) can be easily confused with the binary classification function p.

**Other Strengths And Weaknesses:**

Strengths:
S1. This paper introduces a novel counterfactual explanation framework for temporal graph neural networks on continuous-time dynamic graphs, addressing a clear gap in dynamic graph interpretability.
S2. This paper presents a well-defined methodology that combines Monte Carlo Tree Search with spatio-temporal and gradient-based selection policies.
S3. The experimental evaluation is comprehensive. The authors test the proposed method CoDy on multiple datasets and compare against both factual and greedy baselines.

Weaknesses:
W1. The spatio-temporal and gradient-based policies may introduce a large bias, favoring recent or proximal events while neglecting older, causally critical events.
W2. While the authors claimed that CoDy is model-agnostic, its reliance on prediction logits may produce inconsistent results for TGNNs with differing architectures, which the paper does not investigate.
W3. Discussion on the reason why CoDy outperforms factual methods in achieving better AUFSC- scores is not given. This is a bit counter-intuitive because CoDy is not configured to achieve this.
W4. Although the evaluation is extensive, it relies primarily on quantitative metrics. It would be great if the authors could also incorporate user studies or human-centered evaluations to assess the interpretability of the generated counterfactuals.
W5. Nit picking: the experiments focus on social/interaction networks. Performance on dynamic graphs with fundamentally different dynamics (e.g., biological networks, sensor networks) remains unverified, raising questions about method universality.

**Questions For Authors:**

Please refer to the weaknesses.

**Relation To Broader Scientific Literature:**

This paper advances prior works in two key areas: GNN explainability and counterfactual reasoning. Existing GNN explanation methods focus on static graphs, lacking effectiveness to handle temporal dependencies. For dynamic graphs, prior methods are limited to factual explanations, which identify contributing features but fail to explore "what-if" scenarios. CoDy bridges this gap by adapting counterfactual explanation to temporal contexts, leveraging insights from causal reasoning and temporal graph modeling.

**Theoretical Claims:**

No theoretical claims were made in this paper.

---

> ### Author Rebuttal · Authors · 2025-04-01
>
> We sincerely thank you for your time and insightful feedback on our paper. Please find our rebuttal to the weaknesses below:
>
> **W1: Potential bias in selection policies**
>
> The selection policies are indeed designed as heuristics to introduce a bias, guiding the search efficiently within the vast combinatorial space of potential counterfactual subgraphs (past events). This guidance is crucial for tractability. Our experiments explicitly demonstrate the value of this bias.
> While these policies prioritize recent/proximal/high-impact events, the MCTS framework in CoDy, particularly the exploration component in the selection score (Section 4.3), inherently encourages exploring less-immediately-promising paths. This helps mitigate the risk of exclusively focusing on recent events and allows CoDy to discover counterfactuals involving older or less obviously connected events if they are indeed critical. The search doesn't only consider these events; it prioritizes them while retaining the ability to explore alternatives.
>
> **W2: Reliance on prediction logits**
>
> We appreciate the reviewer's point regarding the term "model-agnostic." We use this term to indicate that CoDy does not require access to the internal architecture, parameters, or gradients of the TGNN model, unlike gradient-based or model-specific explainers. It interacts with the model solely through its input (perturbed graph history) and output (prediction score).
> We assume that a TGNN outputs a continuous output signal (like logits or probabilities), which is the case for all TGNNs that we are aware of. In a case of a TGNN that strictly produces binary outputs, there would indeed be little guidance for efficient search, likely necessitating brute-force approaches (however, we do not percieve this a realisitc setup of a TGNN).
>
> **W3: CoDy outperforms factual methods on $AUFSC_-$ scores**
>
> This is an excellent observation, and we agree it appears counter-intuitive since CoDy optimizes for counterfactuals ($fid_+$). The high $fid_-$/$AUFSC_-$ scores (sufficiency) achieved by CoDy, especially compared to T-GNNExplainer in several settings (Table 1), stem primarily from CoDy's search objective and fallback strategy. CoDy seeks the minimal necessary set of events to change the prediction. When a true counterfactual is not found within the search budget (either because none exists in the constrained search space or the search limit is reached), CoDy's fallback strategy (Section 4.3.5) returns the perturbation set $s_opt$ that caused the largest shift in the prediction logits. This $s_opt$ represents the set of most influential past events identified in the search. Even if not enough to flip the prediction, these events often still capture sufficient factual evidence for the prediction of the model. In fact, our evaluation shows, that this is highly effective in uncovering factual evidence. In contrast, factual methods like T-GNNExplainer aim directly for sufficiency but might identify larger subgraphs or be affected by approximations (as noted in their paper and our Section 5.3.2), potentially leading to lower $fid_-$/$AUFSC_-$ scores in some cases compared to CoDy's focused, impact-maximizing search (even in fallback).
>
> **W4: Lack of user studies or human-centered evaluations**
>
> We fully agree with the reviewer that user studies are invaluable for assessing the practical interpretability and utility of explanations. However, the primary contribution of this work is methodological: introducing the first counterfactual explanation framework (CoDy) specifically for TGNNs on CTDGs and rigorously evaluating its quantitative performance against existing factual methods and a novel greedy baseline using established metrics ($fid_+$, $fid_-$, sparsity, etc.). Given this focus, we considered extensive human-centered evaluations to be beyond the scope of this initial paper. Acknowledging the importance of user studies, we will explicitly recommend them as a direction for future work in the conclusion section of the revised manuscript.
>
> **W5: Experiments focus on social/interaction networks**
>
> Our choice of datasets (Wikipedia, UCI-Messages, UCI-Forums) was motivated by their prevalence in TGNN research and their representation of important real-world dynamic systems. While they primarily fall under social/interaction networks, they exhibit considerable diversity:
> - Unipartite (UCI-Messages) vs. Bipartite (Wikipedia, UCI-Forums).
> - Presence (Wikipedia) vs. Absence (UCI) of edge features.
> - Varying temporal dynamics and graph densities (Table 2).
>
> We believe these datasets provide a strong and representative initial validation of CoDy's effectiveness. While our datasets focus on interaction networks, the methodology extends to broader CTDG applications, such as financial modeling and dynamic recommendation systems.

---

> > ### Comment · Reviewer_5zq2 · 2025-04-05
> >
> > I thank the authors' response, which solved all my previous concerns. I would like to keep my rating.

---

### Official Review · Reviewer_NKwF · 2025-03-08

**Overall Recommendation:** 5

**Summary:**

The paper introduces CoDy, a method for generating counterfactual explanations for Temporal Graph Neural Networks (TGNNs). Unlike existing methods that focus on factual explanations, CoDy explores how minimal modifications to a dynamic graph can alter predictions (counterfactuals).

CoDy employs Monte Carlo Tree Search (MCTS)-based algorithm with heuristic selection policies that leverage temporal, spatial, and local gradient (*why did you call it like this?*) information to efficiently identify counterfactual subgraphs. To benchmark its performance, the authors develop GreeDy - although there are other static counterfactual explainers in the literature - and propose an evaluation framework assessing necessity and sufficiency of explanations. Experiments on three real-world datasets (Wikipedia, UCI-Messages, UCI-Forums) and two TGNN models (TGN, TGAT) show that CoDy outperforms factual and counterfactual baselines.

The results highlight that counterfactual explanations offer a distinct interpretability advantage by illustrating what minimal changes would lead to different outcomes, especially for misclassified predictions. CoDy provides a structured approach to exploring decision boundaries in dynamic graphs.

## Update after rebuttal
For the AC: I changed my score from weak accept to strong accept. I went back and forth with the reviewers, and they've answered most of my questions and followed my suggestions. There were some disagreements with them, but that shouldn't undermine this paper's acceptance. I'm willing to champion this paper as it's the best in my batch of reviews.

**Claims And Evidence:**

The authors claim to be the first to propose a continuous temporal graph counterfactual explainer, *which is fair enough*. To the best of my knowledge only [1] does temporal graph counterfactual explanations on a snapshot-based graph. However, I'm not sure why the authors need to distinguish between the two? The snapshot-based graph methods can be easily transformed into their continous version by just not framing the graph in $\Delta$ time intervals of the same time span.

[1] Prenkaj et al. Unifying Evolution, Explanation, and Discernment: A Generative Approach for Dynamic Graph Counterfactuals. KDD'24

The authors also claim to have a $6\times$ improvement in fidelity. However, I can't seem to find where this claim is supported. Table 1 doesn't showcase this, or am I missing something?

**Essential References Not Discussed:**

I came across this paper [1] that trates temporal graph counterfactual explainability when considering the time-graph as a sequence of snapshots - i.e., what the authors named DTDGs. I believe it should be treated at least in the related work section. I'm not sure wheather it would fit the authors' comparisons since they discretize the time component and get snapshots of the graph changes. I read Prenkaj et al.'s paper and I believe they explain GNNs on a specific snapshot, particularly the first one, and then use their explainer to generatively classify the incoming graphs and explain them dynamically. This doesn't seem to fit with what CoDy is trying to do; however, I still believe that this comparison between "real" TGNN  and snapshot-based counterfactual explainability should be treated in the **Related Work** section. To the best of my knowledge, this is the only work - after some search - in dynamic counterfactual explainability in graphs. Maybe instead of **GreeDy**, the authors could compare against **GRACIE** [1]?

*[1] Prenkaj B, Villaizán-Vallelado M, Leemann T, Kasneci G. Unifying Evolution, Explanation, and Discernment: A Generative Approach for Dynamic Graph Counterfactuals. In Proceedings of the 30th ACM SIGKDD Conference on Knowledge Discovery and Data Mining 2024 Aug 25 (pp. 2420-2431).*

I believe Prado-Romero et al's bibtex citation should be the following.

*@article{prado2024survey,
  title={A survey on graph counterfactual explanations: definitions, methods, evaluation, and research challenges},
  author={Prado-Romero, Mario Alfonso and Prenkaj, Bardh and Stilo, Giovanni and Giannotti, Fosca},
  journal={ACM Computing Surveys},
  volume={56},
  number={7},
  pages={1--37},
  year={2024},
  publisher={ACM New York, NY}
}*

I had complaints in other venues from reviewers that the paper isn't only an arXiv- Rather it is an ACM CSUR publication. Just to give the authors a heads-up.

---

The first paragraph in the related work section misses a lot of static counterfactual explainers (e.g., [2-8] among others).

[2] Abrate & Bonchi. Counterfactual graphs for explainable classification of brain networks. KDD'21

[3] Ma et al. Clear: Generative counterfactual explanations on graphs. NeurIPS'22

[4] Numeroso & Bacciu. Meg: Generating molecular counterfactual explanations for deep graph networks. IJCNN'21

[5] Prado-Romero et al. Robust stochastic graph generator for counterfactual explanations. AAAI'24

[6] Lucic et al. Cf-gnnexplainer: Counterfactual explanations for graph neural networks. AISTATS'22

[7] Bajaj et al. Robust counterfactual explanations on graph neural networks. NeurIPS'21 (*factual-based explainer just like PGExplainer*)

[8] Chen et al. D4explainer: In-distribution explanations of graph neural network via discrete denoising diffusion. NeurIPS'23

[9] Faber et al. Contrastive Graph Neural Network Explanation. ICLM'20 Workshop (*this guy always produces correct counterfactuals, but it's bound by the dataset: i.e., can't generate something ex-novo*)

**Experimental Designs Or Analyses:**

I believe the authors didn't explore enough time-graph datasets. I found out that the TUDataset website has a lot of them that are easy to integrate in the authors' codebase and contain continuous time events. Examples of datasets follow:
1. DBLP-Coauthors [1]
2. BTC-Alpha, BTC-OTC [2]
3. Bonanza [3]

[1] Benson et al. Simplicial closure and higher-order link prediction. Proceedings of the National Academy of Sciences. 2018

[2] Kumar et al. Rev2: Fraudulent user prediction in rating platforms. In WSDM'18

[3] Derr et al. Balance in signed bipartite networks. In CIKM'19

----
The authors decide to adapt a single explainer, PGExplainer, from static scenario to a dynamic one. How? Why only PGExplainer? Also are you adapting it for counterfactuality? If not, how are you evaluating fidelity on it?

Additionally, there are a lot of other counterfactual - although PGExplainer is factual (see question #6) - explainers. For example, Prenkaj et al. [4] adopted a lot of static counterfactual explainers: BDDS [5], MEG [7], CLEAR [6]. The authors might scrutinize the goodness of these adapted explainers in a DTDG scenario. Nevertheless, to be fair to previous work, I would suggest to follow the same strategy to complete the analyses made in this paper. For now, the authors just compare to 2 explainers. Also, in the literature, researchers compare against a random baseline - and I'm not talking about *GreeDy-rand.* - to check the sanity of their proposed method.

[4] Prenkaj et al. Unifying Evolution, Explanation, and Discernment: A Generative Approach for Dynamic Graph Counterfactuals. In KDD'24

[5] Abrate & Bonchi. Counterfactual graphs for explainable classification of brain networks. In KDD'21

[6] Ma et al. Clear: Generative counterfactual explanations on graphs. NeurIPS'22

[7] Numeroso and Bacciu. Meg: Generating molecular counterfactual explanations for deep graph networks. IJCNN'21

---
Recall that fidelity is a measure based on the underlying predictor. See Prado-Romero et al.'s [8] definition (see pag. 23):
$$\Psi(x, x') = \chi(x) − \mathbf{1}[\Phi(x') = y_x],$$ where $x$ is the input, $x'$ is the counterfactual produced. $\chi(x)$ gives 1 if $x$ was correctly classified, and 0 otherwise, and $y_x$ is the ground truth label of $x$. A value of 1 entails that both the explainer and predictor are working correctly. 0 and −1 describe something wrong with the explainer or the oracle. However, we cannot attribute the incorrect function to
the explainer or the oracle. This is a shortcoming of fidelity since it bases the assessment of the correctness on the ground truth label $y_x$ instead of the prediction $\Phi(x)$. Therefore, the literature [1-8], use also validity (correctness) to evaluate their counterfactual's goodneess. This metric is never used in this paper.

Furthermore, the structural distance, measured by Graph Edit Distance (GED) [8], is never used. The used *sparsity* metric just measures the distance on the node features. However, since the authors advocate for spatial distance as well, it's wierd that GED isn't there, which undermines the claims of the search tree producing "close-by" counterfactuals.

[8] Prado-Romero et al. A survey on graph counterfactual explanations: definitions, methods, evaluation, and research challenges. CSUR'24.

---
The authors claim that their local-gradient CoDy variant is the best-informed counterfactual explainer. Table 1 shows that the choice for CoDy is random. Sometimes, even the GreeDy variant surpasses it. That's why correctness (validity) and GED should be here. **Moreover, the abstract states that the authors have a 6$\times$ improvement in fidelity than SoTA methods. This improvement doesn't appear to emerge from the table!**

---
It seems very wierd that the authors try to evaluate both necessity and sufficiency, as does CF$^2$ [9], but never compare to it. Why choose PGExplainer as comparison and not CF$^2$ undermines the authors claims to be the best in *AU FSC-*. This method is specifically developed to minimize both necessity and sufficiency.

[9] Tan et al. Learning and evaluating graph neural network explanations based on counterfactual and factual reasoning. In WWW'22

**Methods And Evaluation Criteria:**

They partially do, however the authors miss a lot of related work and metrics used in the literature to solidify their empirical claims for CoDy. Please see *Experimental Designs and Analyses*.

**Other Comments Or Suggestions:**

No.

**Other Strengths And Weaknesses:**

See other sections, please; the reviewer form is already cluttered with a lot of sections. The paper raises a lot of questions, specifically in the experimental section, and the lack of prudent evaluation with the plethora of static graph counterfactual explanations, and the usage of adequate metrics to measure the goodness of the proposed method. While the authors want to showcase *fidelity-* and *fidelity+* as CF$^2$ [1] does, it's a puzzle why they refrain from comparing CoDy to an adapted version of CF$^2$ to dynamic graphs...

[1] Tan et al. Learning and evaluating graph neural network explanations based on counterfactual and factual reasoning. In WWW'22

**Questions For Authors:**

1. So is CoDy only able to perform edge addition/removal operations to generate the counterfactuals? There's no node addition/removals right?
2. Why are you calling your selection strategy *local-gradient*? It's just a logit-based difference... Are you saying that you have access to the gradients of the predictor? I don't think so; otherwise, you're "blowing-up your cover" for treating black-box TGNNs; rather they should be called grey-boxes. I'd argue that you change this nomenclature to something more suitable.
3. Instead of having the heuristic guide your tree traversal, have you thought of learning the path that guides you toward the counterfactual? If so, how would you do that? If not, why didn't you? CoDy's tree algorithm "screams" reinforcement learning to me: did you consider doing this? My question would be, what's the motivation behind Monte-Carlo Search and not other strategies?
4. In Eq. (7) why is the exploration and exploitation governed by $\alpha$ and $\beta$, and not only $\alpha$? In other words, why isn't the equation equal to $selscore(n_k) = \alpha\cdot score_k + (1-\alpha)\cdot score_{explore}(n_k)$?
5. When you have multiple counterfactual candidates, you state that you select the one with the largest shift in the TGNN prediction. Doesn't this, in some sense, suggest that you're overshooting the decision boundary of your predictor? See Wachter et al's distance requirement [1]. Or is this distance tackled during the search algorithm?
6. The authors use PGExplainer as a baseline for static graphs and adapt it to dynamic ones. I wonder why the authors chose a factual explainer in a counterfactual fashion? This doesn't make sense. How are you evaluating fidelity on a factual explainer? How are the authors generating counterfactuals with a factual explainer? Are you simply removing the factual explanation generated from the input graph to hopefully produce a valid counterfactual?
7. Since fidelity has been shown to have inherent weaknesses as a metric (see [2]), why don't you use correctness (validity) to assess the goodness of the produced counterfactuals?
8. Can you please show the Graph Edit Distance (GED) [2] of your counterfactuals? This is a necessary metric to support your claims of the search tree producing "close-by" counterfactuals.
9. Where is your claim of $6\times$ improvement in fidelity supported? I can't see it from Table 1.
10. I don't understand what correct and wrong predictions are in Table 1. Can you please clarify? I can't seem to find anything related to it in the experimental section.
11. In Fig. 9, it is concerning that Cody-Rand. has >0.35 Jacard similarity with the other "more informed" variants, especially with "local-grad." How do you explain this? Is the local-grad. variant that informed? Recall that is just the difference between logits in the original prediction and the counterfactual one.


[1] Wachter S, Mittelstadt B, Russell C. Counterfactual explanations without opening the black box: Automated decisions and the GDPR. Harv. JL & Tech.. 2017;31:841.

[2] Prado-Romero MA, Prenkaj B, Stilo G, Giannotti F. A survey on graph counterfactual explanations: definitions, methods, evaluation, and research challenges. ACM Computing Surveys. 2024 Apr 9;56(7):1-37.

**Relation To Broader Scientific Literature:**

This works opens doors to graph counterfactual explainability in time-graphs although this field has already been partially explored in [1]. I can't seem to find why [1] nor this paper should or can have a broader scientific impact. Both paper's evaluations are in controlled environments that do not show how they could be employed in real-world scenarios, although both authors argue that *"their datasets are real-world*.

[1] Prenkaj et al. Unifying Evolution, Explanation, and Discernment: A Generative Approach for Dynamic Graph Counterfactuals. In KDD'24

**Theoretical Claims:**

There are no theoretical claims.

---

> ### Author Rebuttal · Authors · 2025-04-01
>
> Thank you for your comprehensive review and valuable feedback.
>
> **1.** CoDy strictly removes events from the input graph reflecting the causal relationship between those events and the prediction; i.e., if events $\\{e_i,...\\}$ would not have happened, the prediction would be different. The counterfactuals thus depend on the identified events, including edge and node additions/removals (note that the evaluated TGNNs only cover additions)
>
> **2.** We use "local-gradient" to reflect the impact that removing a single event has on the prediction logits. The term "local gradient" reflects this in the sense of the influence that an infinitesimal change to the input (removal of event) has on the output. We agree that the term could be misconstrued as implying access to model internals and are thus renaming it to “local-event-impact”.
>
> **3.** Our search algorithm balances exploration and exploitation, dynamically adapting the search based on previously explored trajectories, similar to Monte-Carlo Tree Search. This flexibility is especially important in a vast combinatorial space as in CTDGs.
> We have indeed considered learning an MLP to inform a selection strategy, however, our initial experimentation with this gave a strong indication that such a surrogate fails to perform sufficiently to provide a good selection indication in this setting.
>
> **4.** We have followed your suggestion and simplified the formulation.
>
> **5.** Our goal in selecting the candidate with the largest shift is to identify the minimal set of features whose removal has the most decisive impact on the prediction. In many counterfactual explanation methods (e.g., Wachter et al.), the distance from the decision boundary is a primary metric, and our approach follows a similar principle by prioritizing the “critical” events. To ensure a balanced selection, we combine this approach with a sparsity criterion, avoiding unnecessary complexity while maintaining interpretability.
>
> **6.** Our procedure for PGExplainer was to generate factual explanations - i.e., the subgraph that supports the prediction - and then construct a counterfactual by removing this subgraph from the input graph. Although this is a heuristic adaptation, it provides a baseline for comparing fidelity.
> Adapting most static CF methods (especially generative ones like CLEAR or optimization-based like MEG) to the CTDG setting requires significant methodological changes (handling event sequences, temporal dynamics, defining valid edits in time) and engineering effort, making it a substantial research project in itself. Our focus was on establishing a baseline within the CTDG CF setting, which motivated the development of GreeDy. We believe GreeDy serves as a more direct and fair baseline for CoDy's search strategy within the same problem formulation.
> On a similar note, CF² requires applying continuous masks on nodes/edges which is difficult to translate to the event-based nature of TGNNs like TGAT and TGN.
>
> **7 + 10.** Fidelity definitions vary across works. We follow GraphFramEx’s model-focused fidelity definition (Amara et al. 2022) and clarify this in Appendix D.
>
> We evaluate instances in which the TGNN makes correct predictions separately from instances where it misclassifies future links. We state this in 5.1 under “Explained Instances”. Given your feedback, we have added further clarification on this and the fidelity metric.
>
> **8.** We appreciate the suggestion to use GED. However, CoDy only performs event removal. In this specific setting, the number of removed events (captured by the numerator in sparsity) is the graph edit distance from the original graph's event set to the perturbed graph's event set (where edits are event deletions). Since CoDy operates solely via event removal, sparsity naturally captures structural distance, making GED an unnecessary addition in this context.
>
> **9.** Thank you for highlighting this. The 6.11x improvement in fidelity refers to average improvement of CoDy-spatio-temporal over PGExplainer shown in the table in Appendix F. The number is an artefact of a previous revision and we will replace it with an insight from Table 1 and add an explanation.
>
> **11.** We believe the Jaccard similarities are a natural consequence of the fact that, regardless of the selection policy, all variants start with the same underlying search tree and balance exploration & exploitation, steering the search along promising paths. When the key causal events are strongly influential, even a random policy may discover similar substructures. For the local-gradient variant, although its ranking metric is based on the logit difference, the fact that its explanations significantly overlap with those from the spatio-temporal variant reinforces that the same vital events are consistently identified.
>
> We thank the reviewer for highlighting other related works. We have added GRACIE to related work. While addressing a different problem, it is an important addition for completeness.

---

> > ### Comment · Reviewer_NKwF · 2025-04-03
> >
> > Q1: Nice, thank you for the clarification.
> >
> > Q2: I agree that the new nomenclature makes more sense.
> >
> > Q3: I'd be very interested if you can extend your Monte-Carlo approach to an RL-based approach in the future. I believe that the exploration and exploitation of an RL can be easily adapted to your case.
> >
> > Q4 (**+ additional questions**): Oh, cool! How is now the contirbution of each of the two components? You definitely can measure it now, right? What $\alpha$ did you choose here? Is it the old $\alpha$? If so, how do your empirical insights (experiments and other stuff) change? Do you maintain the same performance?
> >
> > Q5: I'm a bit confused. I read your answer 5 times now, and can't seem to wrap my head around it. Sorry. As a follow-up for me to understand, you said that you use sparsity as a metric to choose the counterfactual that has the lowest impact in the predictor's outcome. How do you measure this impact? And, is sparsity defined as in [1] (pag. 22)?
> >
> > Q6: Ok, my intuition was correct about PGExplainer, pheww :-). Out of curiosity how would you proceed to "port" CF$^2$ in your scenario?
> >
> > Q7 (**fidelity doubts again**): I'm aware that Amara et al. 2022 got already published (in a NeurIPS workshop), however their claims completely ignore citing Tan et al.'s [3] (WWW'22 held 6 months before NeurIPS'22) paper that introduce $fid_+$ and $fid_-$ as scores. *I believe citing the original contribution aka CF$^2$ is a good compromise here*. In my opinion, Yuan et al.'s [4] survey that Amara refers to is completely done for factual explanations and doesn't have anything to do with counterfactuality. In fact, it never talks about fidelity in the entire survey since it doesn't make sense to measure fidelity in factual scenarios. Therefore, you refering to Amara's paper which in turns refers to Yuan's factual explainability survey - where again fidelity is never mentioned - doesn't make sense. My suggestion would be to cite [1,2,3] (one of them).
> >
> > Q10: Can you please let me know what you changed exactly?
> >
> > Q8 (**sparsity confusion, again!**): What are you refering to sparsity here? Graph sparsity? It's a bit confusing since sparsity is a metric defined on the changes in the node features of a graph (see Q5).
> >
> > Q9: Cool!
> >
> > Q11: You mentioned a random perturbation strategy might have a similar effect on the Jaccard similarity. In light of this, can you try a random strategy (e.g., iRand [5] or any other for that matter) in your scenario and report the Jaccard similarity, please? I'm curious to see what the effect of the selection strategy is in reality.
> >
> >
> > [1] Prado-Romero et al. A survey on graph counterfactual explanations: definitions, methods, evaluation, and research challenges. ACM Computing Surveys. 2024 Apr 9;56(7):1-37.
> >
> > [2] Guidotti R. Counterfactual explanations and how to find them: literature review and benchmarking. Data Mining and Knowledge Discovery. 2024 Sep;38(5):2770-824.
> >
> > [3] Tan et al. Learning and evaluating graph neural network explanations based on counterfactual and factual reasoning. WWW'22
> >
> > [4] Yuan et al. Explainability in graph neural networks: A taxonomic survey. TPAMI 2022 Sep 5;45(5):5782-99.
> >
> > [5] Prado-Romero et al. Are Generative-Based Graph Counterfactual Explainers Worth It?. InJoint European Conference on Machine Learning and Knowledge Discovery in Databases 2023 Sep 18 (pp. 152-170). Cham: Springer Nature Switzerland.

---

> > > ### Author Response · Authors · 2025-04-04
> > >
> > > Thank you for your continued engagement and follow-up questions.
> > >
> > > Q4: We changed the formula as suggested. To maintain consistency with our original experiments, we set the new $\alpha=\frac{2}{3}$, preserving the original 2:1 weighting between exploration and exploitation terms ($\alpha=2$, $\beta=1$).  Because this change maintains the same balance, **all empirical results, metrics, and insights remain unchanged.**
> > > The contribution of the two terms cannot directly be measured, as the terms interact throughout the search. However, the sensitivity analysis presented in Appendix G provides insights into the impact of this balance, demonstrating the impact of adjusting this balance.
> > >
> > > Q5:
> > > **1. Sparsity:** We define the sparsity of an explanation as $sparsity=\\frac{|\\mathcal{X}\_{\\varepsilon_i}|}{C(\\mathcal{G},\\varepsilon_i,k,m_{max})|}=\\frac{Number\\ of\\ events\\ in\\ explanation}{Number\\ of\\ candidate\\ events}$ (see Appendix D), reflecting the sparsity of event selection. In other words, mapping to the definition in [1], we define $x=C(\\mathcal{G},\\varepsilon_i, k, m_{max})$ (set of candidate events considered for explanation) and $x'=C(\\mathcal{G},\\varepsilon_i,k,m_{max})\\setminus\\mathcal{X}\_{\\varepsilon_i}$ (candidate events without events identified as part of explanation). $D_{inst}(x, x')$ then captures the number of events in the explanation ($|\\mathcal{X}\_{\\varepsilon_i}|$). We choose the number of candidate events as denominator as it represents the maximum perturbation size within the search space, providing a consistent scale for comparison across instances. Using the entire graph history would result in extremely small and less informative sparsity values.
> > >
> > > **2. Explanation Selection:** For selecting a counterfactual example from multiple identified examples, we first filter the identified samples to keep only those with the minimum size (minimum $|\\mathcal{X}\_{\\varepsilon_i}|$), directly minimizing the number of changes (event removals) aligning with the "closest possible world" in the words of Wachter et al.. If and only if there remain multiple counterfactuals with the same minimal size, we then use the magnitude of the prediction shift (largest $\\Delta(p_{orig},p_j)$, Eq. 6) as tie-breaker, selecting the most decisive explanation among equally minimal options.
> > >
> > > Q8: Adding to Q5: In short, for our case: $GED=|\mathcal{X}\_{\\varepsilon_i}|$, which is the numerator in our sparsity definition. Thus, the sparsity captures the structural distance ($GED$), normalized by the size of the search space ($|C(...)|$).
> > >
> > > Q6: In theory, the main hurdle in porting CF² lies in implementing a continuous edge (and feature) masking mechanism into the explained TGNNs. In a TGN(-attn) model this could be done by integrating such masks into the temporal attention layer. Importantly, one would have to account for correctly handling memory updates.
> > > Practically, computational cost are a big hurdle. CF² requires fitting the explanation model (the masks) for each explained instance. Given that inference in complex TGNNs like TGN is highly computationally expensive (as shown by runtime analysis in Appendix F), iterating through a training/optimization process involving repeated TGNN calls for every single explanation would be prohibitively slow for evaluation and practical application on large dynamic graphs.
> > >
> > > Q7: Thank you for your diligence regarding the citation history for fidelity. We agree that acknowledging the foundational work is essential and will cite the original contribution (CF²). While fidelity concepts are broadly used in GNN explaination literature (e.g., Yuan et al. [4], Sec. 7.2.1), the specific formulation and application to counterfactual necessity and sufficiency are clearly articulated in CF².
> > >
> > > Q10: 1. We replace "wrong" with "incorrect". 2. In *5.1 Explained Instances* we add: "we evaluate instances where the TGNN makes correct predictions separately from instances where it makes incorrect predictions." 3. We clarify *5.2 Fidelity* in line with Q7. 4. We add more detailed descriptions to the formulas in Appendix D.
> > >
> > > Q11: To clarify: CoDy-random is not a purely random strategy. It utilizes the complete search framework (score based selection, simulation, expansion, backpropagation). The "random" aspect applies mainly to guiding the exploration of unvisited nodes. This inherent guidance explains why CoDy-random finds explanations overlapping with those of more informed policies, particularly if certain events are highly influential. For completeness, we've run an experiment with iRand, obtaining the following Jaccard Similarity scores compared to CoDy (same settings as Appendix F.3):
> > > - CoDy-random: 0.08
> > > - temporal: 0.08
> > > - spatio-temporal: 0.09
> > > - local-event-impact: 0.09
> > >
> > > These low scores (<0.1 vs >0.35 among CoDy variants in Appendix F.3) confirm that a purely random strategy differs significantly, highlighting the effectiveness of CoDy even with a random selection policy.

---

### Official Review · Reviewer_Sw8d · 2025-03-12

**Overall Recommendation:** 3

**Summary:**

This paper proposes CoDy and GreeDy, two algorithms that can generate counterfactual explanations for continuous-time dynamic graphs (CTDGs). The main component of the algorithm is a Monte-Carlo-Tree Search to find good edge combinations for removal. The method is experimentally evaluated on three dynamic graphs.

**Claims And Evidence:**

The main claim of this paper is the method CoDy. While some evidence for the method is there, I think it could be made more compelling (see below).

**Essential References Not Discussed:**

I am not aware of highly relevant missing references.

However, I think the connection between discrete time and continuous time graphs could be more deeply discussed. There are simple processing methods to convert discrete into continuous time graphs and vice-versa.
However, for discrete time graphs, there are already explanation methods, like Prenkaj et al. (2024). I wonder wheter applying these methods on a discretized version of the graph would not already solve the problem. I think this connection necessitates further consideration.


Prenkaj, Bardh, et al. "Unifying Evolution, Explanation, and Discernment: A Generative Approach for Dynamic Graph Counterfactuals." Proceedings of the 30th ACM SIGKDD Conference on Knowledge Discovery and Data Mining. 2024.

**Experimental Designs Or Analyses:**

The claimed superiority over the baselines is studied in experiments with several datasets and baselines. I could not find any substantial flaws in the setup.

However, I think the results are mixed so I am not entirely enthusiastic.
First, the generated explanations do not seem to be very reliable. As far as I understand, e.g., the char score is supposed to be between 0 and 1. On average, the best-performing methods presented in this work show scores between 0.4 and 0.5., which does not seem very good. GreeDy further shows very good performance on TGAT in particular, so there is no clear suggestion on which method to use.

The spatio-temporal method has the best performance, but also makes the most calls to the model (Table 5, Appendix).
Ablations of the different components: The method has several steps, and I am unsure how systematic the exploration of these is. So far, I only see an ablation for the selection policy. There is some improvement over random, but it is small. Further, the results from GreeDy show that greedy search is also quite good. Given the 10-fold increase in runtime (Appendix F.1), I am not convinced that the full method is worth the additional investment in practice.

Generally speaking, the experimental design is valid, but the experimentation could be more extensive, and have better insights (it is basically a huge results table, where some methods will on some datasets, some on otheres, but I cannot see a clear pattern or learning from it).

**Methods And Evaluation Criteria:**

I think the metrics are okay, but I wonder how the sparsity of the explanation can be selected with CoDy/GreeDy. It seems the evaluation AUFSC was done with different sparsity levels. I there a hyperparameter to select this or did the authors just rank the explanations? I am not sure whether this metric would be valid then, because if a method assigns non-sparse but always correct explanations it should get a high score, vs. a method that tries to optimize sparsity, but is not always correct.

I think it would make more sense to compute this curve for a hyperparameter that trades off sparsity vs. fidelity / correctness.

There are three datasets used, which is okay but also not a very extensive benchmarking.

**Other Comments Or Suggestions:**

Minor
*  i.e., (comma missing, l.124, right)
*  Equation 2 does not seem to make much sense: What does the vector notation mean? It seems we have a vector where one element (upper) is a graph and the other one (lower) is an integer

**Other Strengths And Weaknesses:**

**Summary.** I find it hard to form an opinion about this paper. On the one hand there are no obvious methological flaws. The writing is good. However, I am not very enthusiastic about the experimental results, which don't seem to tell a clear story. Also the problem feels very specific. Overall, I  award this paper a borderline rating.

**Questions For Authors:**

Can you please explain how the sparsity of the explanations is controlled and how the AUFSC is computed? Also, what sparsity levels are the fid+, fid- scores computed at? How do you explain the difference in performance between the algorithm versions?

**Relation To Broader Scientific Literature:**

This work contributes a new method for *continuous-time dynamic graph counterfactual explainability*. I agree that there are not many methods for this graph counterfactual of problem and therefore the paper does make a contribution.

However, as the many specifiers to the domain suggest, I think the problem is relatively niche and only relevant to a very specific subgroup of the ICML community.

**Theoretical Claims:**

There are no theoretical claims in this work.

---

> ### Author Rebuttal · Authors · 2025-04-01
>
> We sincerely thank you for the detailed and insightful review.
>
> **Runtime vs. Performance Trade-off**
>
> The noted potential 10-fold increase in model calls for CoDy is largely a configurable design choice, not an inherent limitation of CoDy. CoDy, in our experiments, continues searching after finding the first counterfactual (CF) to identify potentially sparser explanations. GreeDy terminates immediately upon finding a CF or reaching a local optimum.
> As shown in Appendix F.2, CoDy typically finds the first counterfactual explanation relatively quickly, often within a number of iterations comparable to GreeDy's total calls. If configured to stop after finding the first CF (maintaining fidelity while deprioritizing minimal sparsity), CoDy's runtime becomes much closer to GreeDy's, while still being more robust. Thus, CoDy’s runtime is highly adjustable, allowing users to trade off speed for sparsity and more interpretable explanations as needed. We will clarify this in the paper.
>
> **Sparsity Control**
>
> Neither CoDy nor GreeDy take sparsity as an input; they identify the minimal necessary event set to flip the prediction. Sparsity is an outcome of the search, not a hyperparameter. T-GNNExplainer also determines size automatically; only PGExplainer is configured with a fixed sparsity of 0.2. The fidelity values $fid_+$ and $fid_-$ are computed aggregately over all explanations found for the test instances, representing the overall success rate for the method across its generated explanations. To calculate the AUFSC, we integrate over the cumulative fidelity achieved at sparsity levels (the fraction of instances for which a method found a counterfactual explanation with $sparsity \leq s$, at sparsity level s). This is a standard way to evaluate methods producing variable-size outputs, directly comparing their ability to find concise and necessary explanations.
>
> **Connection to Discrete-Time Methods**
>
> While CTDGs can be discretized, this process inherently loses temporal precision, as multiple events are grouped within fixed intervals, erasing exact timing and ordering information. TGNNs designed for CTDGs (e.g., TGN, TGAT) explicitly leverage this fine-grained information. Applying a DTDG explanation method to a discretized CTDG explains a different, approximated system and model, not the original CTDG and TGNN. Methods like Prenkaj et al. (2024) are designed for such models that have different assumptions about graph evolution. CoDy directly explains higher-fidelity CTDG models, ensuring explanations faithful to the underlying temporal structure.
>
> **Performance**
>
> We agree that char scores around 0.5-0.6 are not perfect, but they represent a significant achievement for this novel and challenging task. As the harmonic mean of $fid_+$​ and $fid_−$, $char$ is sensitive to the lower of the two. Finding minimal necessary counterfactuals (high $fid_+$) on complex temporal graphs is inherently difficult, especially for correctly predicted instances where substantial evidence might support the original prediction (Section 5.3.1). Achieving these scores demonstrates a meaningful ability to identify crucial subgraph structures.
>
> We appreciate the reviewer noting GreeDy's strong performance. We introduce GreeDy precisely as a competitive baseline. While GreeDy is effective when its greedy path aligns well with the optimal solution, CoDy's approach offers greater robustness against local optima. Appendix F.2 illustrates this: CoDy's performance continues to improve with more iterations, eventually surpassing GreeDy even in challenging settings.
>
> We respectfully disagree that there are no clear patterns. The results consistently show: (i) Counterfactual methods (CoDy/GreeDy) significantly outperform factual baselines on $fid_+$, as expected. (ii) Spatio-temporal and Local-gradient selection policies substantially outperform Random and often Temporal policies (Section 5.3.4). (iii) There's a notable difference in explainability between correct and wrong predictions (Section 5.3.1).
> To make the results easier to interpret, we will reorganize the results tables and group evaluation together based on if the predictions were correct or wrong, showing a clearer picture of the results.
>
> **Relevance & Significance**
>
> Regarding the perceived specificity of our topic, we emphasize that CTDGs offer high-fidelity representations for ubiquitous evolving systems like financial networks, social media, and recommendation platforms, where powerful TGNNs are increasingly applied. As these TGNNs operate in high-stakes domains, understanding their 'black-box' decisions through interpretable methods is critical for trust, regulatory compliance, and debugging; counterfactual explanations provide particularly actionable insights in these contexts. Therefore, CoDy addresses the crucial and growing need for trustworthy TGNNs on high-fidelity CTDGs, bridging a significant gap for deploying these models responsibly in real-world applications.

---

> > ### Comment · Reviewer_Sw8d · 2025-04-04
> >
> > I thank the authors for their responses. I better understand the sparsity/fidelity scores now.
> >
> > Regarding the experimental results, I see that there are some base finding, i.e., such that the proposed methods outperform random selection. But these are not the most insightful to me. I am particularly interested in comparing CoDy vs. GreeDy, also to give practitioners a guideline on which to chose and with which policy. Overall, I think the presentation of the results should be improved.
> >
> > While the rebuttal has improved my understanding, I think my main points are still valid and I maintain my score.

---

> > > ### Author Response · Authors · 2025-04-04
> > >
> > > We thank the reviewer for the continued involvement and feedback. We'd like to share how we are revising the paper for a camera-ready version to give a clearer presentation of the results.
> > >
> > > **1. Enhancing presentation of results**
> > >
> > > We reorganize the results in Table 1 and group them by explanations for correct predictions (+) and incorrect predictions (-) to enhance clarity. To exemplify this improvement, see the reorganized section on the $char$ metric, showing a clearer pattern:
> > >
> > > | | uci-m (+) | uci-f (+) | wiki (+) | uci-m (-) | uci-f (-) | wiki (-) |
> > > | --- | --- | --- | --- | --- | --- | --- |
> > > | GreeDy-rand | 0.04 | 0.08 | 0.09 | 0.14 | 0.12 | 0.19 |
> > > | GreeDy-temp | 0.22 | 0.50 | 0.17 | 0.49 | 0.47 | 0.45 |
> > > | GreeDy-spa-temp | *0.31* | *0.53* | 0.23 | 0.54 | 0.46 | 0.54 |
> > > | GreeDy-gradient | 0.18 | 0.39 | 0.14 | 0.51 | 0.42 | 0.43 |
> > > | CoDy-rand | 0.19| 0.43 | 0.24 | 0.52 | 0.47 | 0.58 |
> > > | CoDy-temp | 0.23 | 0.49 | 0.22 | 0.55 | *0.54* | 0.62 |
> > > | CoDy-spa-temp | **0.31**  | **0.54**  | **0.30** | *0.57* | 0.52 | *0.65* |
> > > | CoDy-gradient | 0.27 | 0.50 | *0.27* | **0.58**  | **0.57**  | **0.68** |
> > >
> > > Additionally, we are revising the text accompanying the results to go beyond just reporting numbers by adding interpretation as to why certain methods/policies excel in specific scenarios (e.g., linking CoDy-local-gradient's success on wrong predictions to its ability to exploit model sensitivities when it errs; linking CoDy-spatio-temporal's success on correct predictions to leveraging structural context)
> > >
> > > **2. Directly comparing GreeDy and CoDy**
> > >
> > > We explicitly acknowledge that while GreeDy performs well in some settings (validating it as a strong baseline), CoDy's strength lies in its robustness and adaptability. GreeDy's success often depends on the initial greedy path aligning well with the optimal solution, which isn't guaranteed. CoDy's search framework allows it to explore more broadly and backtrack, making it less susceptible to these pitfalls, leading to better or more reliable explanations, particularly as evidenced by its stronger performance with more iterations (Appendix F.2) and across different policies. The revised presentation will make this trade-off (GreeDy's speed vs. CoDy's robustness) much clearer.
> > >
> > > **3. Add paragraph on practical implications**
> > >
> > > We add the following paragraph on practical implications:
> > >
> > > In practice, CoDy presents the most robust and performant choice. While potentially requiring more computation if searching for minimal sparsity, its performance is more reliable across diverse scenarios. By choosing to terminate the search upon finding the first valid counterfactual, a practical implementation can significantly reduce runtime while retaining CoDy's robust search capabilities. On the other hand, if computational efficiency and rapid explanation generation are paramount, GreeDy presents a compelling, faster alternative. It can yield good results quickly, particularly if its initial greedy choices happen to align with an effective counterfactual path. However, this efficiency comes with a higher risk of suboptimal explanations and the potential of ending-up in local optima.
> > > The spatio-temporal selection strategy is best fit as selection strategy in real-world applications where ground-truth labels are not available. It performs best on correct predictions, which are most common in properly functioning TGNNs, and only slightly worse than the local-gradient policy in explaining incorrect predictions.

---

### Decision · Program_Chairs · 2025-05-01

**Decision:**

Accept (poster)

**Comment:**

The paper introduces a method for generating counterfactual explanations for temporal graph neural networks.

There were significant discussions between authors and the reviewers and also among the reviewers. I believe Reviewer NKwF, an expert in the area, played a crucial role in convincing us to provide an acceptance recommendation.

The reviewers unanimously liked the paper. In particular, the reviewers appreciated the framework for continuous temporal graph counterfactual explainer. While they were minor concerns, such as char scores of around 0.5-0.6, I believe this would be a good starting point to ignite future progress on this problem. I encourage the authors to add a discussion about this in the limitation section. There was also some related work (mainly about the connection between discrete and continuous time) that has not been cited, and I encourage the authors to include these references in the camera-ready version. Also, Reviewer NKwF provides a lot of excellent points during the discussion, and the authors have provided many clarifying answers, which I think can be included in the paper to clarify the significance of the work more clearly.